# Endosomal chloride/proton exchangers need inhibitory TMEM9 β-subunits for regulation and prevention of disease-causing overactivity

Rosa Planells-Cases [1,9], Viktoriia Vorobeva [1,2,9], Sumanta Kar [1,9], Franziska W. Schmitt [1,3], Uwe Schulte [4,5], Marina Schrecker[6], Richard K. Hite [6], Bernd Fakler [4,7] & Thomas J. Jentsch [1,8] ✉

The function of endosomes critically depends on their ion homeostasis. A crucial role of luminal $Cl^-$, in addition to that of $H^+$, is increasingly recognized. Both ions are transported by five distinct endolysosomal CLC chloride/proton exchangers. Dysfunction of each of these transporters entails severe disease. Here we identified TMEM9 and TMEM9B as obligatory β-subunits for endosomal ClC-3, ClC-4, and ClC-5. Mice lacking both β-subunits displayed severely reduced levels of all three CLCs and died embryonically or shortly after birth. TMEM9 proteins regulate trafficking of their partners. Surprisingly, they also strongly inhibit CLC ion transport. Tonic inhibition enables the regulation of CLCs and prevents toxic $Cl^-$ accumulation and swelling of endosomes. Inhibition requires a carboxy-terminal TMEM9 domain that interacts with CLCs at multiple sites. Disease-causing *CLCN* mutations that weaken inhibition by TMEM9 proteins cause a pathogenic gain of ion transport. Our work reveals the need to suppress, in a regulated manner, endolysosomal chloride/proton exchange. Several aspects of endosomal ion transport must be revised.

The function and trafficking of endosomes and lysosomes critically depends on ion transport across their membranes[1]. Most work on endosomal ion homeostasis has focused on luminal pH, but other ions, such as $Cl^-$ also play pivotal roles[2]. Traditional thinking focused on an indirect role of electrogenic $Cl^-$ *transport* in facilitating luminal acidification by shunting currents of $H^+$ ATPases. More recent data point to an important role of luminal $Cl^-$ *concentration*[3–6] which influences vesicular volume and the activity of luminal enzymes. Major players in endosomal $Cl^-$ transport are CLC chloride/proton exchangers[7]. The importance of endolysosomal ion transport is evident from pathologies associated with their dysfunction. With CLC $2Cl^-/H^+$ exchangers, these include neurodegenerative syndromes[8–11], lysosomal storage, osteopetrosis and albinism[12,13], and proteinuria and kidney stones[14].

The CLC family of $Cl^-$ channels and transporters comprises nine members[7]. Members of the first homology branch, ClC-1, -2, -Ka, and -Kb, function as $Cl^-$ channels at the plasma membrane. Members of the second (ClC-3 to ClC-5) and third branches (ClC-6 and ClC-7) function as $2Cl^-/H^+$-exchangers on intracellular vesicles (hence vCLCs). ClC-3 to ClC-6 are found in overlapping endosomal compartments, whereas ClC-7 is mainly expressed on lysosomes[7]. vCLCs function in

[1]Leibniz-Forschungsinstitut für Molekulare Pharmakologie (FMP), Berlin, Germany. [2]Graduate Program of the Free University Berlin, Berlin, Germany. [3]Graduate Program of the Humboldt University Berlin, Berlin, Germany. [4]Institute of Physiology, Faculty of Medicine, University of Freiburg, Freiburg, Germany. [5]Logopharm GmbH, March-Buchheim, Breisgau, Germany. [6]Structural Biology Program, Memorial Sloan Kettering Cancer Center, New York, NY, USA. [7]Signalling Research Centres BIOSS and CIBSS, Freiburg, Germany. [8]NeuroCure Cluster of Excellence, Charité Universitätsmedizin Berlin, Berlin, Germany. [9]These authors contributed equally: Rosa Planells-Cases, Viktoriia Vorobeva, Sumanta Kar. ✉e-mail: Jentsch@fmp-berlin.de

endolysosomal Cl⁻ accumulation and acidification[7]. Although all vCLCs are 2Cl⁻/H⁺-exchangers and share, with the exception of ClC-6[15], similar voltage- and pH-dependencies[7], the pathologies associated with their dysfunction differ. This might be explained by different expression patterns or regulation.

CLCs operate as dimers with one ion translocation pathway per subunit[7]. All CLCs function as homodimers, but heterodimers within the same homology branch are possible[16,17]. Several CLC proteins bind ancillary "β-subunits". Barttin[18] and Ostm1[19] are obligatory β-subunits for ClC-K channels and ClC-7 exchangers, respectively, and GlialCam[20] is a facultative β-subunit of ClC-2. However, the mechanisms by which they regulate the function of their CLC partners are obscure. It has remained unknown whether ClC-3 to ClC-6 have β-subunits.

Here we show that endosomal ClC-3, ClC-4, and ClC-5 require TMEM9 and TMEM9B for normal function and protein stability. These obligatory β-subunits regulate both the trafficking and ion transport of their CLC partners via distinct carboxy-terminal domains. Strong constitutive suppression of Cl⁻/H⁺ exchange by TMEM9/9B allows to markedly increase ion transport even with minor relaxation of this inhibition. Human *CLCN* mutations interfering with this control lead to toxic gain of endosomal ion transport and disease.

## Results

### TMEM9 proteins are obligatory β-subunits of endosomal CLCs

Membrane fractions from mouse brain were investigated by high-resolution complexome-profiling via csBN-MS, a technique combining native gel electrophoresis with quantitative mass spectrometry[21] (Fig. 1a). The resulting abundance-mass profiles reflected functional homo/hetero-dimeric assemblies (profile-peaks at ~400 kDa), as well as incompletely assembled CLC monomers (~200 kDa). Our data uncovered perfect overlap, indicative for co-assembly, of the ClC-3/4 profiles with those of TMEM9 and TMEM9B, type I transmembrane proteins of ~20 kDa. A similar overlap was detected with ClC-5 which is poorly expressed in brain[7].

Multi-epitope affinity-purifications from mouse brain combined with mass spectrometry (meAP-MS) confirmed effective co-assembly of the TMEM9 proteins with ClC-3 to -5, indicated by their tight co-clustering in two-dimensional t-SNE plots[22] (Fig. 1b,c). The MS-derived abundance and ratio data indicated specific and stoichiometric co-assembly of ClC-3 to -5 with TMEM9 and TMEM9B.

TMEM9 and TMEM9B are ~56% identical. Orthologs are present in probably all animals, with mammals having two isoforms (Supplementary Fig. 1). TMEM9 proteins display a cleavable signal peptide, a glycosylated amino-terminus, a single transmembrane span, and a cytosolic carboxy-terminus of ≈70 residues. Both TMEM9 (referred to as TMEM9A or T9A) and TMEM9B (T9B) were reported to reside in lysosomes and late endosomes[23–26] and to have functions in cellular signaling and cancer[24,25,27].

T9A co-immunoprecipitated two different ClC-3 splice variants[28], ClC-4 and ClC-5, but not ClC-7, from lysates of transfected cells (Fig. 1d). To determine which parts of T9 proteins interact with CLCs, we constructed chimeras with Ostm1, the ClC-7 β-subunit which shares the same transmembrane topology[19]. Co-immunoprecipitations revealed that the transmembrane domain (TMD) of T9A was necessary to bind ClC-5 (Fig. 1e). Expression databases (www.proteinatlas.org/) indicate almost ubiquitous expression of *T9A* and *T9B* transcripts. Western blots confirmed wide-spread expression of either protein while revealing differences in expression levels (Fig. 2a). Together with the differential distribution of their CLC partners[7], this suggests that ratios of particular CLC/T9 complexes vary between tissues.

We disrupted both *Tmem9* genes singly and in combination in mice. Both *Tmem9⁻/⁻* (*T9a⁻/⁻*) and *Tmem9b⁻/⁻* (*T9b⁻/⁻*) mice lacked obvious phenotypes. For *T9a⁻/⁻* mice, this agrees with previous work[24], while *T9b⁻/⁻* mice have not yet been reported. Crossing both strains

failed to generate live *T9a⁻/⁻*/*T9b⁻/⁻* offspring in >30 litters, with lethality of double KO mice occurring after E12.5 (see Methods). Hence T9 proteins have crucial physiological functions. Both isoforms may partially replace each other.

### Mutual dependence of CLC and T9 proteins

Although *T9a* or *T9b* disruption did not change transcript levels of *Clcn3 – Clcn5* or *T9b* and *T9a*, respectively (Supplementary Fig. 2), T9A protein levels were upregulated upon loss of T9B (Fig. 2b). Reciprocal dependencies of CLC and T9 protein levels were analyzed in brain, which abundantly expresses ClC-3 and ClC-4, and in kidney, the main site of ClC-5 expression. As *T9a⁻/⁻*/*T9b⁻/⁻* mice were not viable, we examined tissues from single KOs or from mice homozygous for disruption of one isoform and heterozygous for the other (Fig. 2b). Loss of T9 subunits differentially affected levels of their CLC partners. Most striking was the strong decrease of ClC-4 with loss of either T9A or T9B in kidney. Renal expression of ClC-5 was dependent on T9B, but not on T9A. By contrast, levels of ClC-3 and of the non-interacting ClC-6 or ClC-7 exchangers appeared unchanged. Cultured fibroblasts derived from *T9a⁻/⁻*/*T9b⁻/⁻* embryos had almost completely lost ClC-3, -4 and -5 proteins (Fig. 2c). The abundance of ClC-7 and several endolysosomal marker proteins was unchanged.

Conversely, T9 proteins depend on CLCs for stability (Fig. 2d-f). Most prominent was the massive reduction of T9A protein in *Clcn3⁻/⁻* brain. It may be explained by the concomitant reduction of ClC-4, which is observed in brain and other tissues of *Clcn3⁻/⁻* mice[29], and by low ClC-5 levels in brain. This leaves only small amounts of CLC partners for T9A stabilization. T9B was unchanged in *Clcn3⁻/⁻* kidney and only moderately reduced in brain. T9A levels also depended on ClC-4 and ClC-5 (Fig. 2e,f). T9B lost its glycosylation in *Clcn5⁻/⁻* kidney (Fig. 2f), suggesting that it may not reach the Golgi without ClC-5. Accordingly, disruption of *Clcn5* abolished T9A and T9B expression in apical endosomes of renal proximal tubules where these proteins normally co-localize (Fig. 2g,h). T9B appeared non-glycosylated in brain which almost lacks ClC-5 (Fig. 2d,e). The proportion of glycosylated T9B seemed increased in *Clcn3⁻/⁻* kidney (Fig. 2d), where ClC-3 and ClC-4 now compete less with ClC-5 for T9B binding. Preferential ClC-5/T9B interaction was further supported by the observation that loss of T9B, but not of T9A, reduced ClC-5 amounts in kidney (Fig. 2b).

### Predominant endosomal localization of T9A and T9B

As binding partners of endosomal ClC-3, -4 and -5, T9A and T9B should localize to the same compartments as their CLC partners[7]. Antibodies against C-terminal epitopes of either T9 isoform gave punctate cytoplasmic staining in HeLa cells which only sparsely co-localized with early endosomal rab5 or late endosomal and lysosomal lamp2 (Supplementary Fig. 3a,b). The sparse co-localization might be owed, in part, to a shielding of T9 epitopes in CLC/T9 complexes (see below). Transfection of the rab5 mutant Q79L enlarges endosomal compartments to which we had previously[30] localized ClC-5. Endogenous T9A and T9B were found on these structures (Supplementary Fig. 3c).

The endocytic apparatus of renal proximal tubules is conveniently stratified. Both T9A and T9B were detected in a subapical, endosome-enriched rim which prominently expresses ClC-5[30–32] (Fig. 2g,h). No significant co-localization was observed with late endosomal/lysosomal ClC-7 or lamp1 (Fig. 2i). Bone marrow-derived macrophages showed unambiguous co-localization of T9B and ClC-5 on large macropinosomes[33] (Fig. 2j).

Localization of T9A and T9B to endosomes agrees with their role as β-subunits of ClC-3 to ClC-5. However, others[24,25] have described T9A and T9B as localizing predominantly to lysosomes. T9A had been reported to bind lysosomal H⁺-ATPases and help with their assembly[24]. Non-ratiometric Lysotracker® staining had suggested a markedly less acidic lysosomal pH (pH_lys) in *T9a⁻/⁻* cells[24]. However, we observed no differences in Lysotracker® or Lysosensor® labeling in

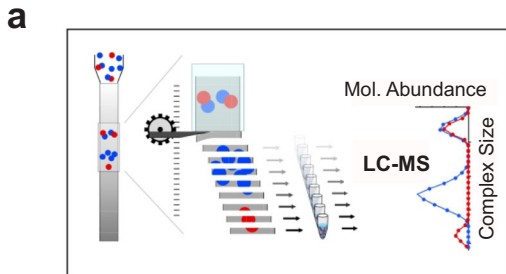

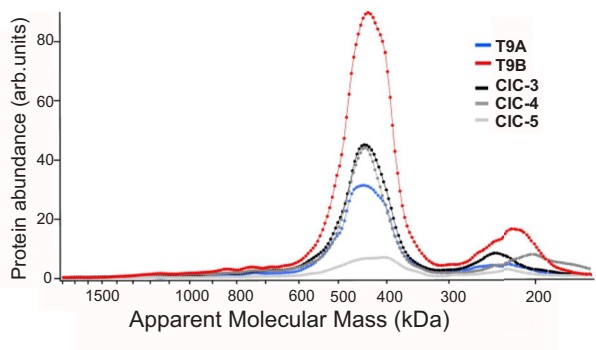

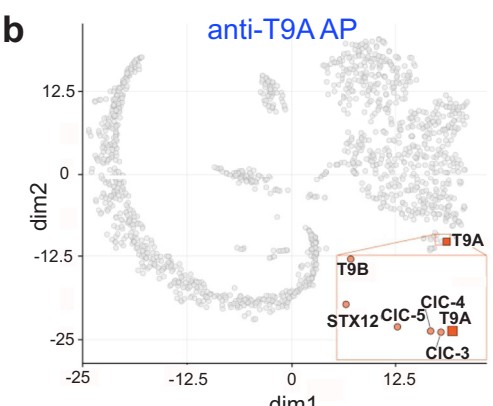

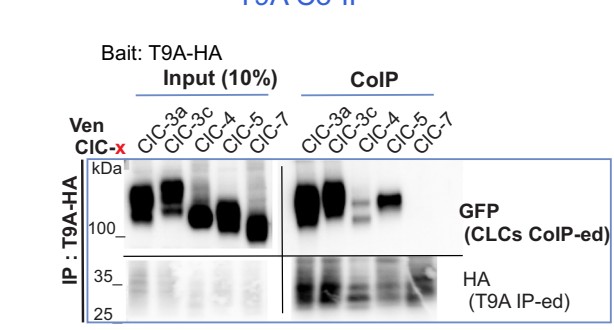

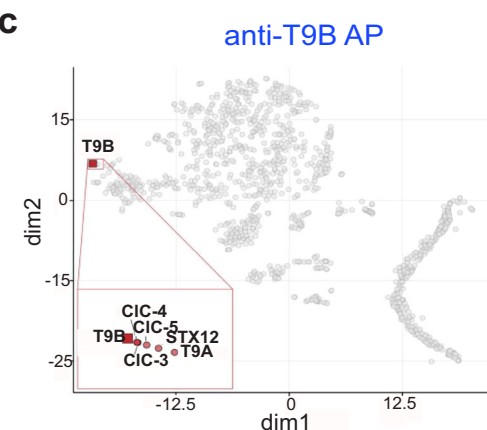

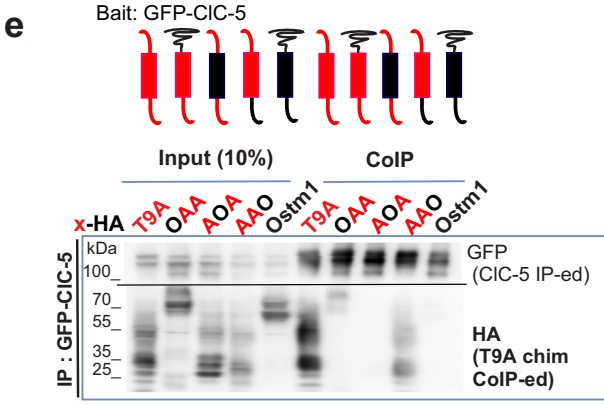

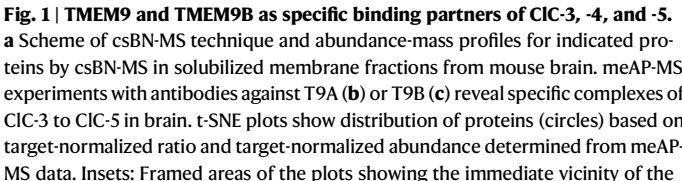

**Fig. 1 | TMEM9 and TMEM9B as specific binding partners of ClC-3, -4, and -5.** **a** Scheme of csBN-MS technique and abundance-mass profiles for indicated proteins by csBN-MS in solubilized membrane fractions from mouse brain. meAP-MS experiments with antibodies against T9A (**b**) or T9B (**c**) reveal specific complexes of ClC-3 to ClC-5 in brain. t-SNE plots show distribution of proteins (circles) based on target-normalized ratio and target-normalized abundance determined from meAP-MS data. Insets: Framed areas of the plots showing the immediate vicinity of the target. Close proximity of the identified proteins indicates consistent and specific co-purification in all APs. **d** T9A-HA co-precipitates Venus-tagged ClC-3, −4, and −5, but not ClC-7, from transfected HeLa cells. **e** Co-IPs of ClC-5 with chimeras between T9A and Ostm1 in transfected HEK cells. Luminal, transmembrane, and cytosolic domains of T9A (indicated by A, red) were replaced with Ostm1 (O, black) as indicated. Similar results observed in three independent co-IP experiments. Unprocessed blots are available in Source Data provided with this paper.

HeLa cells or mouse embryonic fibroblasts (MEFs), and pH$_{lys}$ was undistinguishable between WT, *T9A*$^{-/-}$ and *T9A*$^{-/-}$/*T9B*$^{-/-}$ HeLa cells in ratiometric measurements (Supplementary Fig. 3d-f). Neither complexome profiling nor affinity-purification of T9 complexes (Fig. 1a-c) hinted at an association of T9 with V-type-ATPases.

**TMEM9 proteins regulate vCLC traffic via an acidic cluster**
To investigate whether T9 proteins change the ion transport of vCLCs, we chose ClC-5 which yields the largest plasma membrane (PM) currents[7,34]. ClC-5 currents were abolished by co-expressing either T9A

or T9B in *Xenopus* oocytes (Fig. 3a) or HeLa cells (Supplementary Fig. 4). This is in line with the recent observation[26] that T9B inhibits ClC-3 and ClC-4 PM currents by unknown mechanisms. T9 proteins may have decreased ClC-5 surface expression and/or inhibited its ion transport. Surface expression of a ClC-5 mutant carrying extracellular HA epitopes was strongly reduced by T9A or T9B (Fig. 3b). This does not exclude an additional inhibitory effect on ClC-5 transport activity.

Intracellular retention of ClC-5 might result from an interaction of T9 C-termini with the cytoplasmic sorting machinery[35]. While a large truncation of the T9B C-terminus (ΔC$_{large}$, Fig. 3c) no longer suppressed currents and allowed surface expression of co-expressed HA-

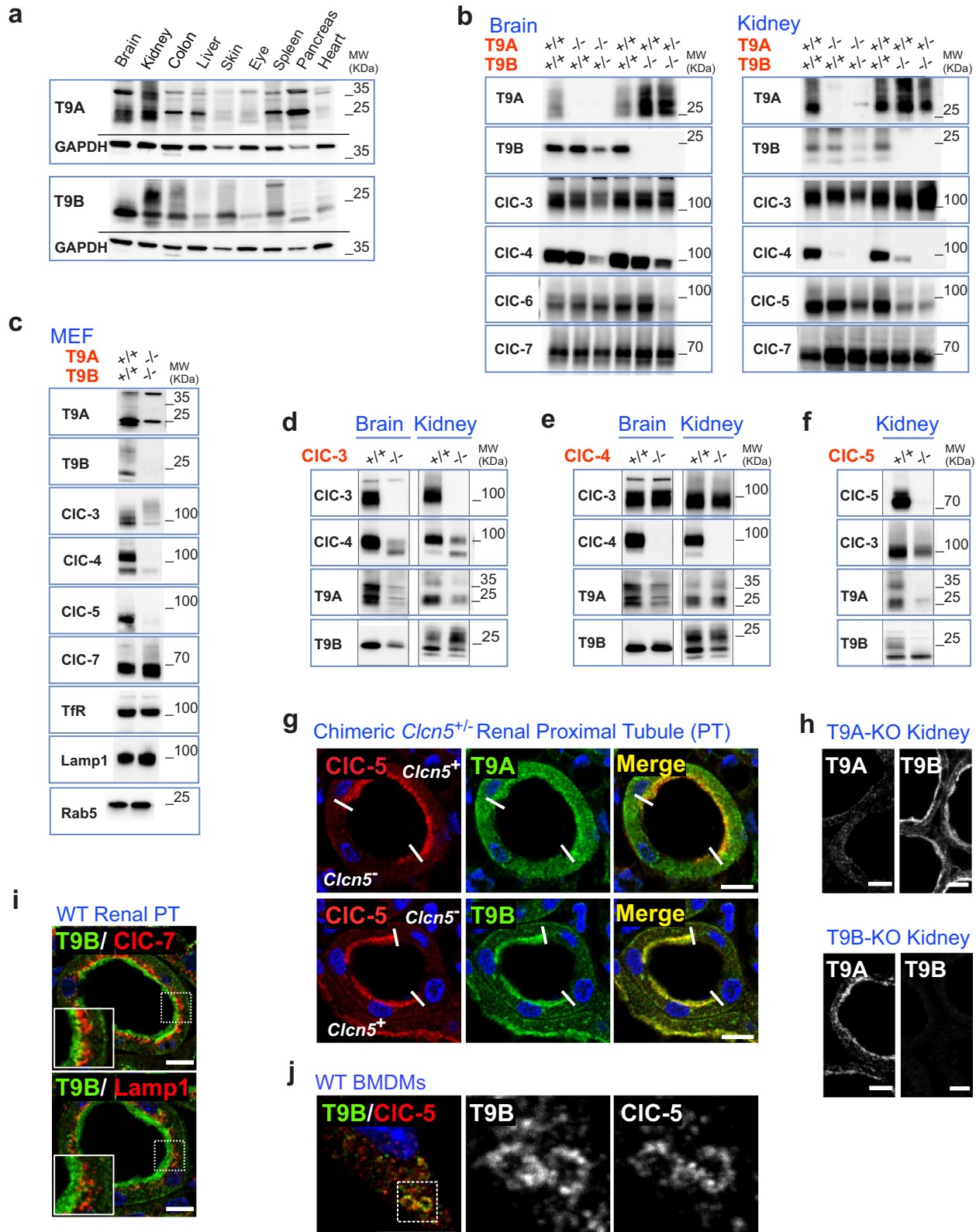

**Fig. 2 | TMEM9 and TMEM9B as obligatory binding partners of ClC-3, -4, and -5.**
**a** Distribution of T9A and T9B proteins in mouse tissues. Similar results obtained in two independent experiments. **b** T9 and CLC protein levels in brain and kidney of mice disrupted for *Tmem9* (*T9A*) or *Tmem9b* (*T9B*). **c** T9, CLC, and marker protein levels in mouse embryonic fibroblasts from WT and *T9a*−/−/*T9b*−/− mice. **d** *Clcn3* KO on ClC-4 and T9 protein levels. **e** *Clcn4* KO on ClC-3 and T9 proteins. **f** *Clcn5* KO on ClC-3 and T9 proteins. **b–f**, similar results observed in three independent experiments. **g** *Clcn5* KO abolishes T9A and T9B expression, in green, in apical endosomes of renal proximal tubules (PT). Chimeric expression of X-chromosomal ClC

−5 (left, red) in *Clcn5*+/− female mice reveals cell-intrinsic mechanism. **h** Specificity of T9 antibodies. Cortical sections of kidneys from *T9a*−/− and *T9b*−/− mice were stained with T9ACT and T9BCT antibodies against T9A and T9B. **i** T9B (green) does not colocalize with lysosomal ClC-7 (red) or lamp1 (red) in mouse renal PTs. Zoomed areas (white box) are displayed in lower left corner. **j** T9B (green) co-localizes with ClC-5 (red) on macropinosomes of BMDMs. Enlarged area is individually displayed for T9B and ClC-5. Similar results were obtained in two independent experiments. Scale bars, 10 μm throughout. Unprocessed blots are available in Source Data provided with this paper.

ClC-5, a smaller deletion ($\Delta C_{small}$, Fig. 3c) suppressed both HA-ClC-5 PM expression and currents (Fig. 3d,e). Further mutants revealed that the stretch deleted in $\Delta C_{int\_prox}$ is responsible for reducing surface expression. Of note, although T9B$^{\Delta Cint\_prox}$ co-expression allowed ClC-5 surface expression, it still suppressed ClC-5 currents.

Although $\Delta C_{int\_prox}$ had similar effects with T9A and T9B, the deleted sequences lack significant homology (Supplementary Fig. 1a). However, they contain conspicuous clusters of negative charges. In their vicinity, we found serines and threonines reported to be phosphorylated (www.phosphosite.org) (T9A: [137]Thr; T9B: [138]Ser, [143]Ser). Proteomic analysis of mouse brain additionally revealed phosphorylation of [122]Tyr and [123]Thr that precede the acidic cluster of T9A (Supplementary Fig. 6). This suggested the presence of *phosphorylated Acidic Clusters* (pACs) that are implicated in endocytic trafficking and Golgi targeting of membrane proteins[35–37]. Similar Ser-, Thr- or Tyr-containing acidic clusters are found in T9 proteins from distant phyla (Supplementary Fig. 1b). Replacing three negatively charged residues by alanine in T9A$^{\Delta pAC}$ and T9B$^{\Delta pAC}$ (Fig. 3c) indeed increased PM expression of co-expressed HA-ClC-5 (Fig. 3f,g) without, however, enabling chloride currents (Fig. 3i). T9A$^{\Delta pAC}$ also augmented PM expression of ClC-3 in HeLa cells (Fig. 3j). Eliminating potentially phosphorylated residues in these clusters moderately increased ClC-5 surface expression with T9A$^{Y122A}$ and T9A$^{T123A}$ (Fig. 3f). With T9B, S138A, but not S143A, enhanced HA-ClC-5 PM expression (Fig. 3g). If ClC-5 and T9 traffic as stable complexes, T9 mutations should similarly affect the PM residence of both proteins. Indeed, epitope-tagged HA-T9B showed increased surface expression with deletions eliminating pAC (Fig. 3e).

### T9 C-terminal inhibitory domains (CID) inhibit vCLC ion flux

Deleting T9 C-termini almost entirely ($\Delta C_{large}$ mutants) allowed surface expression and currents of co-expressed ClC-5, whereas with T9$^{\Delta Cint\_prox}$ or T9$^{\Delta pAC}$, ClC-5 resided at the PM without yielding currents. This suggested that T9 C-termini inhibit ClC-5 ion transport. Interestingly, currents were observed when ClC-5 was co-expressed with T9$^{\Delta pAC}$ mutants with C-terminally added HA-epitopes (Fig. 3h). We hypothesized that the tags might interfere with the recognition of a C-terminal inhibitory domain ('CID').

The ~20 most distal residues of T9 proteins are highly conserved between T9A and T9B (Fig. 3c) and between distant animal phyla (Supplementary Fig. 1a,b). Compatible with a regulatory role, C-terminal T9B residue [190]Ser can be phosphorylated (www.phosphosite.org). Centered on this residue we replaced five amino-acids by alanines in mutants T9A$^{\Delta CID}$ and T9B$^{\Delta CID}$ (Fig. 3c). When combined with mutations allowing PM expression in T9A$^{\Delta pAC,\Delta CID}$ and T9B$^{\Delta pAC,\Delta CID}$, the inhibitory effect on ClC-5 currents was relieved (Fig. 3i). This was also observed with alanine substitutions on both sides of this short stretch (Fig. 3c,i), suggesting that inhibition of ion transport requires a rather long carboxy-terminal domain. T9$^{\Delta pAC,\Delta CID}$ did not change ClC-5 current properties (Fig. 3a,i). Importantly, neither the C-terminal addition of an HA-epitope, nor the $\Delta$CID mutation changed the effect of T9A on surface expression of co-expressed ClC-5 (Supplementary Fig. 7). This suggests that pAC and CID domains specifically affect surface expression and ion transport capacity, respectively, of ClC-5 and other vCLCs.

The majority of ClC-3, -4 and -5 localizes to, and functions in, endosomes[7]. To test for CLC ion transport activity in their native environment, we examined the generation of large vesicles by ClC-3 overexpression[28,38], an effect which is not observed with ClC-4 or ClC-5 (Fig. 4a). Mutations uncoupling Cl$^-$ from H$^+$ transport had previously revealed that vacuolization depends on Cl$^-$/H$^+$ exchange[38]. This dependence was also found with vacuoles induced by pathogenic variants of ClC-6[11,39] or ClC-7[13,40]. We re-examined the effect of E282A *unc*oupling *unc* mutation, which converts ClC-3 into a pure Cl$^-$ conductor, on vacuolization (Fig. 4b). This mutation suppressed, and the

transport-deficient *td* mutation E339A, which almost eliminates both Cl$^-$ and H$^+$ transport[41], prevented vacuolization (Fig. 4b). Hence, vesicle enlargement can be used to assay ion transport by vCLCs.

Vacuoles generated by ClC-3 overexpression were acidified (Fig. 4b)[38]. Although Lysosensor® staining was variable, overall Lysosensor® fluorescence could be used as semi-quantitative readout for large vesicles in flow cytometry (FC) (Supplementary Fig. 8a). Transfection of Venus-ClC-3 into HeLa cells led to a population of cells displaying increased Lysosensor® fluorescence which, confirming the imaging data, was strongly reduced when ClC-3 carried *unc* or *td* mutations (Fig. 4c).

T9A or T9B co-transfection suppressed ClC-3-induced vacuolization (Fig. 4d). This suppression depended on direct protein interactions because it was not observed with Ostm1/T9 chimeras that cannot bind vCLCs (Supplementary Fig. 8b). Suppression of vacuolization depended on T9 C-termini. Co-expressing ClC-3 with T9A$^{\Delta CID}$ or T9B$^{\Delta CID}$ gave large vesicles (Fig. 4e), as did co-expression with T9B$^{\Delta CID\_prox}$, T9B$^{\Delta CID\_dist}$ and T9B$^{\Delta CID\_xdist}$ mutants (Supplementary Fig. 8c), confirming the extension of CID as determined by electrophysiology (Fig. 3i). T9$^{\Delta CID}$ mutants co-localized with ClC-3 on giant vacuoles (Fig. 4e), indicating that the lack of inhibition is not due to a dissociation of T9$^{\Delta CID}$ proteins from ClC-3.

### Role of T9 CID−vCLC interactions in regulation and disease

The inhibition of CLC ion transport by T9 proteins may involve direct CID-CLC interactions. This notion was supported by activity-dependent antibody accessibility of carboxy-terminal T9 epitopes. In cells co-transfected with Venus-ClC-3 and T9A, ClC-3 was found on vesicular, endosome-like structures. Surprisingly, these structures almost lacked labeling with antibodies against T9 C-termini (Figs. 3j, 4f). T9A was rather found in ER-like structures. However, T9AC2 and T9BC2 antibodies detected T9A and T9B, respectively, on large vacuoles generated by GFP-ClC-3/T9$^{\Delta CID}$ (Fig. 4e). Hence, apparent co-localization of both proteins correlated with ClC-3 disinhibition. We speculated that the epitopes recognized by T9AC2 and T9BC2 antibodies (Supplementary Fig. 1a), although not directly located in CID, might be shielded by CID binding to ClC-3. Indeed, co-transfected N-terminally HA-tagged T9A was detected on small ClC-3-positive puncta by anti-HA (Fig. 4fC), but not by T9AC2 antibodies (Fig. 4fB). Both antibodies detected T9A on the large vacuoles generated by ClC-3/T9A$^{\Delta CID}$ (Fig. 4gB,C). We conclude that CID inhibits ion transport by binding to ClC-3 and that relief from inhibition involves CID/CLC dissociation. Epitope-shielding likely contributes to the difficulties in localizing T9 proteins to defined subcellular structures of native cells (Supplementary Fig. 3a,b).

To find CID binding sites on ClC-3, we were guided by the previous identification of the linker connecting helices J and K as 'receptor' for an inhibitory N-terminal domain of ClC-2[42–44]. Replacing part of the JK linker of ClC-3 by that of ClC-2 in ClC-3$^{JK-C2}$ indeed made the T9 epitope accessible for the T9AC2 antibody (Fig. 4h) and relieved the inhibitory effect of T9A (Fig. 5a,b). Three disease-causing *CLCN4* mutations, R360S, R364G and P369L[45], change JK linker residues. Vacuole generation by ClC-3$^{R418S}$, the equivalent to ClC-4$^{R360S}$, was resistant to T9A inhibition, as was vacuolization by ClC-3$^{K422G}$ ($\approx$ ClC-4$^{R364G}$) (Fig. 5c). No effect on T9A inhibition was found with ClC-3$^{P427L}$ ($\approx$ ClC-4$^{P369L}$). T9B had similar effects as T9A (Supplementary Fig. 9a). This suggested that ClC-4$^{R360S}$ and ClC-4$^{R364G}$ cause disease by a detrimental gain of function.

To search for additional mutants able to relieve T9-mediated inhibition, we analyzed pathogenic *CLCN4* mutations affecting cytoplasmic residues without changing currents. Seven disease-associated mutations in the first CBS domain of ClC-4 mostly give WT currents[45]. When equivalent mutations where inserted into ClC-3, only ClC-3$^{R710T}$ ($\approx$ ClC-4$^{R652T}$) and ClC-3$^{T713V}$ ($\approx$ ClC-4$^{I655V}$) generated giant vesicles in the presence of T9A or T9B (Fig. 5d, Supplementary Fig. 9b). However,

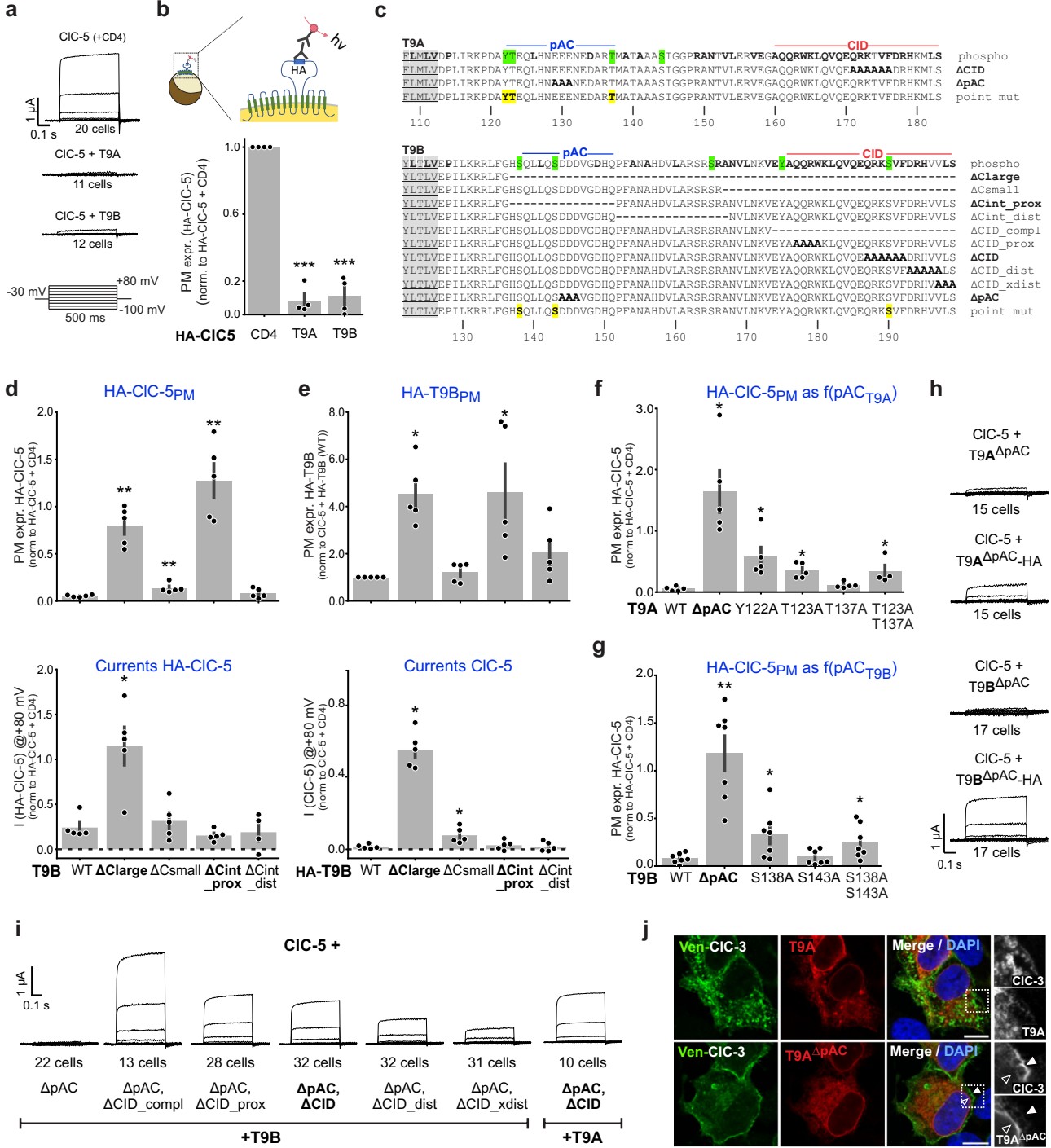

**Fig. 3 | Functional characterization of C-terminal motifs of T9 proteins.**
**a** Averaged background-subtracted currents from oocytes expressing ClC-5 + CD4 (ctrl), T9A or T9B. Below, voltage-clamp protocol. **b** Surface expression of HA-ClC-5 in oocytes is suppressed by co-expression of T9A or T9B. Top, luminescence assay for surface expression. **c** Mutants in T9A and T9B carboxy-termini. Top lines: bold, conserved between isoforms; green, potentially phosphorylated residues. Bottom, singly mutated residues. Left, TMD residues on gray background. **d** Effect of T9B, WT and deletion mutants, on surface expression (above) and currents (below) of co-expressed HA-ClC-5. Currents normalized to mean currents of HA-ClC-5 + CD4 @ +80 mV (0.47 μA). **e** Effect of HA-T9B deletions on surface expression of HA-T9B (above) and currents of co-expressed ClC-5 (below). Currents normalized to mean currents of ClC-5 + CD4 @ +80 mV (3.0 μA). **f, g** Effect of pAC mutants of T9A or T9B, respectively, on HA-ClC-5 surface expression. **h** C-terminal addition of HA-epitopes to T9A^ΔpAC and T9B^ΔpAC weakens their inhibition of ClC-5 currents.

**i** Combining ΔpAC with CID mutations in T9B and T9A enables currents from co-expressed ClC-5. **j** T9A^ΔpAC increases surface expression of ClC-3 in transfected HeLa cells. Filled and open arrowheads indicate plasma membrane and nuclear membrane, respectively. Representative images from technical duplicates and n≥3 biological replicates. Scale bar, 10 μm. Each data point in panel (**b**) (n = 4), (**d**–**f**) (n = 5), and (**g**) (n = 7) represents the mean value from measurements using the same batch of oocytes. Data shown as mean ± SD. Values are compared using non-parametric two-tailed Mann–Whitney test with false discovery rate corrected using the Benjamini–Hochberg procedure. *, $p < 0.05$; **, $p < 0.01$; ***, $p < 0.001$. Data in (**d**–**g**) are compared to ClC-5 + T9B (**d**, **g**), to +HA-T9B (in **e**) or to + T9A (in **f**). Western blots examining the levels of oocyte-expressed proteins (Supplementary Fig. 5) do not change the conclusions from (**d**–**g**). Numerical data and exact $p$ values are given in Source Data provided with this paper.

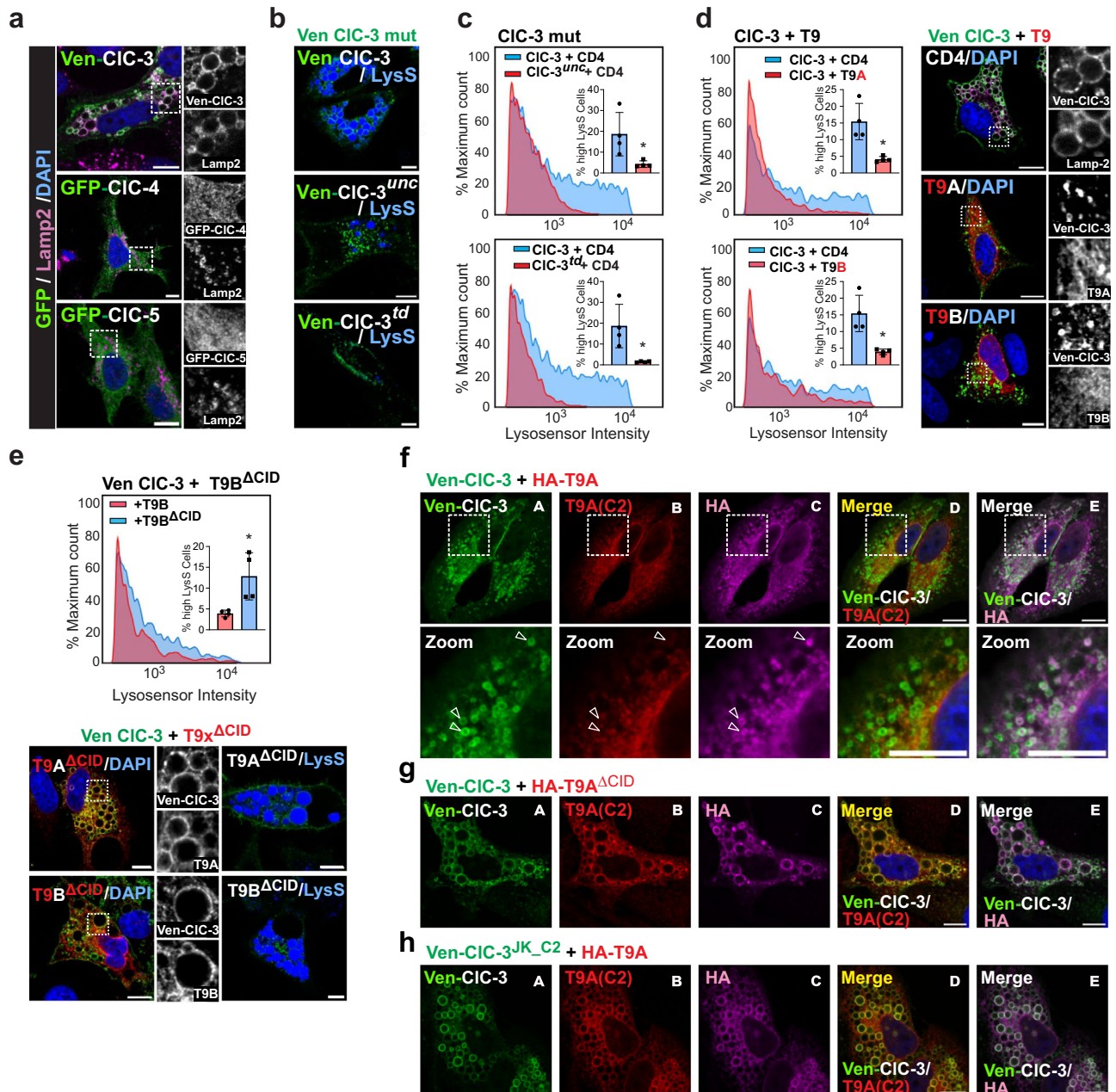

**Fig. 4 | T9 proteins inhibit vacuole generation by ClC-3 overexpression in a CID-dependent manner. a** Large lamp2-positive vacuoles 48 h after HeLa cell transfection with ClC-3, but not ClC−4 or ClC−5 (green). **b** Live cell imaging for Venus fluorescence (green) and acidic pH (blue, using Lysosensor) of cells 48 h after transfection with Venus-ClC-3 (top), Venus-ClC-3$^{unc}$ (center), or Venus-ClC-3$^{td}$ (bottom). Variable Lysosensor staining of vacuoles and inhibition of vacuole formation by *unc* and *td* mutants. **c** Flow cytometry (FC) analysis of Lysosensor reveals cell population with acidified vacuoles in ClC-3 transfected cells, which is reduced with Venus-ClC-3$^{unc}$ and absent with Venus-ClC-3$^{td}$. Same reference for ClC3 + CD4 in both panels. **d** Co-transfection of T9A or T9B suppresses vacuolization as shown by FC (left) and IF (right). Same reference for ClC-3 + CD4 in both panels. **e** The ΔCID mutation in either T9A or T9B suppressed inhibitory effect on vacuolization. Data values in insets of panels c-e represent percentage of cells exhibiting high

Lysosensor intensity from $n = 4$ biological replicates. Data given as mean ± SD. *, $p < 0.05$ using non-parametric two-tailed Mann-Whitney test. Numerical data and exact $p$ values are given in Source Data provided with this paper. Same set of control data (ClC-3 + CD4, blue) was used in upper and lower panels in (**c**). Similarly, same set of control data was used in upper and lower panels of (**d**) (blue). Same data set for ClC-3 + T9B (red) was used in (**d**) (lower panel) and (**e**). **f** T9A is detected on ClC-3-positive puncta (green) in ClC-3/HA-T9A transfected cells by anti-HA (magenta), but not with the T9AC2 (red) antibody (arrowheads). **g** Both anti-HA and T9AC2 antibodies detect T9A on large vacuoles in cells co-expressing HA-T9A$^{ΔCID}$ with ClC-3. **h** Both anti-HA and T9AC2 antibodies detect T9A on large vacuoles in cells expressing HA-T9A with ClC-3$^{JK\_C2}$ ($^{416}$RRRKST to QVMRKQ). Representative images in a, b-h, from technical duplicates and n≥3 biological replicates. Individual channels are depicted from the zoomed area in (**a**, **d**, and **e**). All scale bars, 10 μm.

none of equivalent ClC-5 mutants yielded detectable currents together with T9A$^{ΔpAC}$ (Supplementary Fig. 10c). Hence the degree of disinhibition of ion transport by pathogenic *CLCN4* R652T and I655V mutants and their ClC-3 and ClC-5 equivalents is incomplete. However,

this partial activation of CLC/T9 ion transport suffices to enlarge vesicles and to cause disease.

Mapping the seven ClC-4 CBS1 mutants to the structure of ClC-3$^{41,46}$ revealed that $^{710}$Arg ($^{652}$Arg in ClC-4) and $^{713}$Thr ($^{655}$Ile in ClC-4) are

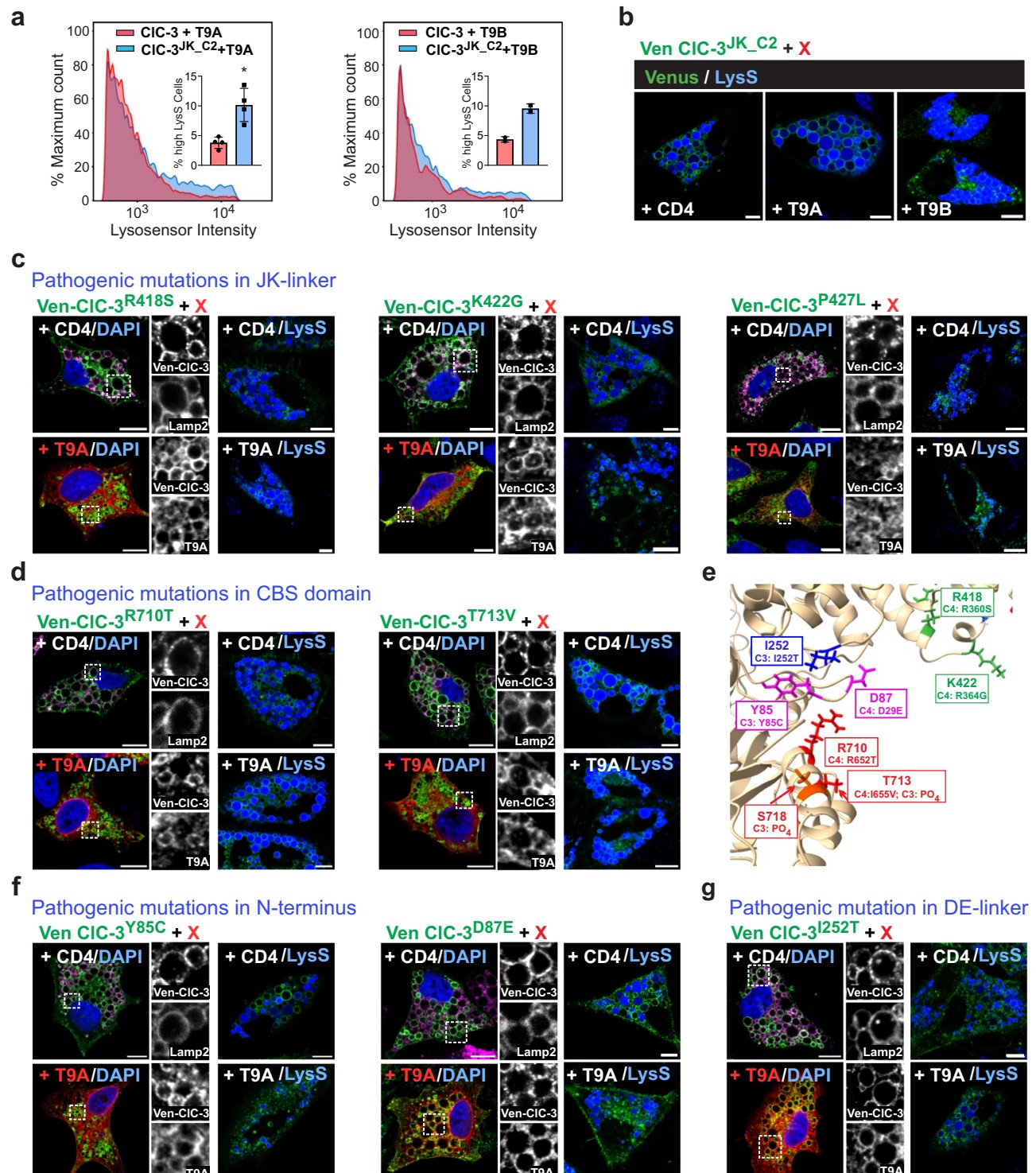

**Fig. 5 | Human pathogenic *CLCN* mutations affecting CLC−T9 interactions.**
**a** Vacuolization by ClC-3$^{JK\_C2}$ is only moderately inhibited by T9A and T9B (Lyso-sensor FC analysis). **b** Resistance of ClC-3$^{JK\_C2}$ (green) to T9A and T9B in individual cells (live cell imaging, Lysosensor, in blue). **c** ClC-3 mutants equivalent to patho-genic *CLCN4* mutations display partial resistance to T9A for R418S and K422G but not for P427L. Effects were weaker than with ClC-3$^{JK\_C2}$ and were hidden in back-ground in FC analysis. **d** Partial resistance to T9A of ClC-3$^{R710T}$ and ClC-3$^{T713V}$ (equivalent to pathogenic *CLCN4* CBS1 mutants). Five pathogenic *CLCN4* mutants, located in the interior of CBS1, remained T9 sensitive (Supplementary Fig. 10). **e** Localization in ClC-3 structure[41,46] of human *CLCN* mutations interfering with T9 inhibition. Amino-acids numbered according to ClC-3b. Lower line in box, human

pathogenic mutations in *CLCN3* (C3) or *CLCN4* (C4) and phosphorylation (PO$_4$) found in database. Color code: Magenta, N-terminus; blue, DE linker; green, JK linker; red, CBS1. **f** T9-resistance of pathogenic *CLCN3* mutation Y85C and of D87E (equivalent to pathogenic *CLCN4* D29E) in amino-terminus. **g** T9-resistance of pathogenic *CLCN3* mutation I252T in DE linker. Data values in insets of (**a**) represent percentage of cells exhibiting high Lysosensor intensity from one experiment from $n = 4$ biological replicates. Data given as mean ± SD. *, $p < 0.05$ using non-parametric two-tailed Mann-Whitney test. Exact $p$ values given in Source Data. Representative images in (**b**−**d**) and (**f**−**g**) obtained from technical duplicates and $n \geq 3$ biological replicates. Split optical channels from zoomed areas shown in (**c**, **d**) and (**f**, **g**). All scale bars, 10 μm.

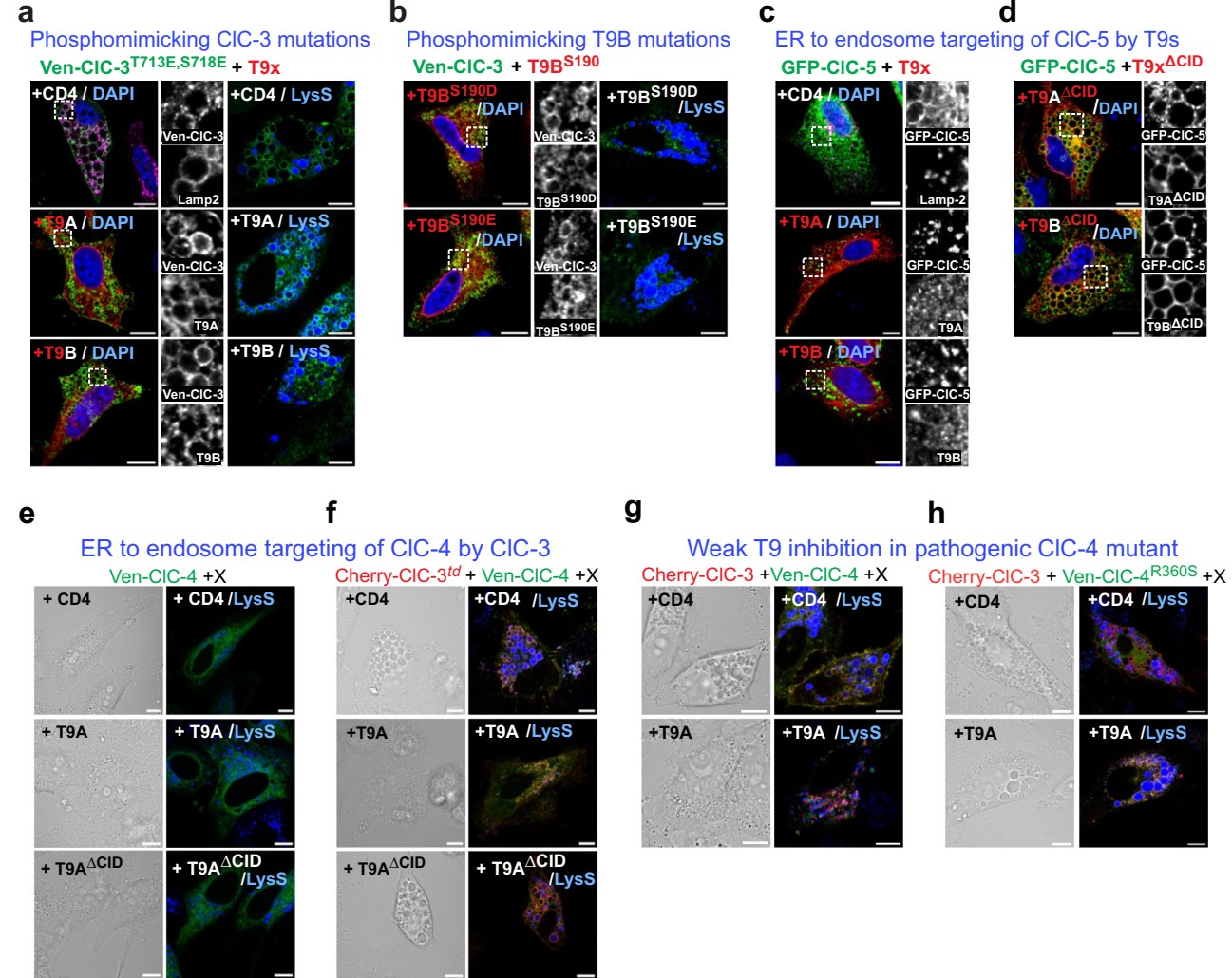

**Fig. 6 | Features of CID-CLC interactions. a** Phosphorylation-mimicking mutations in ClC-3 CBS1 weaken inhibition by T9A and T9B. Left, immunocytochemistry; right, live cell imaging. **b** Phosphorylation-mimicking mutations in T9B $^{190}$Ser weaken inhibition of vacuolization. Immunocytochemistry (left) and live cell imaging (right). **c** T9 co-expression change ClC−5 localization from ER-like (top) to punctate endosome-like (lower panels), shown by immunocytochemistry. **d** T9A$^{\Delta CID}$ or T9B$^{\Delta CID}$ co-expression with ClC-5 induces vacuolization. **e** Neither T9A nor T9A$^{\Delta CID}$ changes ER-like pattern of ClC-4 expression, nor induces vacuolization. **f** Co-expressing transport-deficient ClC-3$^{td}$ with ClC-4 induces vacuolization that is suppressed by T9A but not T9A$^{\Delta CID}$. **g** Co-expressing WT ClC-3 and ClC-4 leads to T9-suppressible vacuolization. **h** Co-expressing WT ClC-3 with disease mutant ClC-4$^{R360S}$ leads to T9-resistant vacuolization. **e**–**h** phase contrast (left) of live cell images for Lysosensor® (right). Representative images in (**b**–**d**) and (**f**, **g**) obtained from technical duplicates and n≥3 biological replicates. Split optical channels depicted from the zoomed areas in (**a**–**d**). All scale bars, 10 μm.

located on the same side of the second α-helix of CBS1 and face the cytosol where they might interact with CID (Fig. 5e). Residues lacking an effect on CID-mediated inhibition are rather buried in the protein (Supplementary Fig. 10a). Roughly between $^{710}$Arg and the JK linker we find three more residues implicated in disease, amino-terminal $^{85}$Tyr and $^{87}$Asp and the DE-linker residue $^{252}$Ile (Fig. 5e). ClC-3$^{Y85C}$ and ClC-3$^{I252T}$ are associated with *CLCN3* neuropathy[9], and ClC-4$^{D29E}$ (≈ ClC-3$^{D87E}$) with *CLCN4* disease[45]. Transfection of either ClC-3$^{Y85C}$, ClC-3$^{D87E}$, or ClC-3$^{I252T}$ elicited T9A- and T9B-resistant vacuolization (Fig. 5f,g, Supplementary Fig. 9c-d), again suggesting that these human mutations cause disease by increasing endolysosomal 2Cl$^-$/H$^+$ exchange.

Databases (www.phosphosite.org) report phosphorylation of ClC-3 $^{718}$Ser and $^{713}$Thr, the equivalent to ClC-4 $^{665}$Ile which is mutated in disease[45] and that was studied above. Both residues are located on the same face of the CBS1 helix (Fig. 5e). Replacing both residues by phosphorylation-mimicking Glu led to T9-resistant vacuolization (Fig. 6a). The database also reports phosphorylation of $^{190}$Ser in the CID

of T9B. Mutating this residue to Glu or Asp led to moderately enlarged vesicles in cells co-transfected with ClC-3 (Fig. 6b). Hence, phosphorylation of either ClC-3 or T9 residues may activate ion transport by weakening CLC/T9 interactions.

Collectively, these results suggest that CIDs of T9 β-subunits inhibit CLC ion transport by interacting with an extended region formed by the second α-helix of CBS1, residues of the amino-terminus, and the DE and JK linkers. Several pathogenic mutants of ClC-3 or ClC-4 cause a toxic gain of function because they are partially resistant to inhibition by T9 β-subunits. Phosphorylation of interacting residues may physiologically regulate vCLC activity.

**Correctly targeted ClC-4 and ClC-5 can cause vacuolization**
The fact that overexpression of ClC-3, but not of ClC-4 and -5, caused vacuolization (Fig. 4a) might result from different subcellular localization. As ClC-5 apparently enabled the transport of T9B to endosomes in kidney (Fig. 2g), we asked whether, conversely, T9B carried ClC-5 to endosomes. T9B co-transfection indeed changed the mainly ER-like

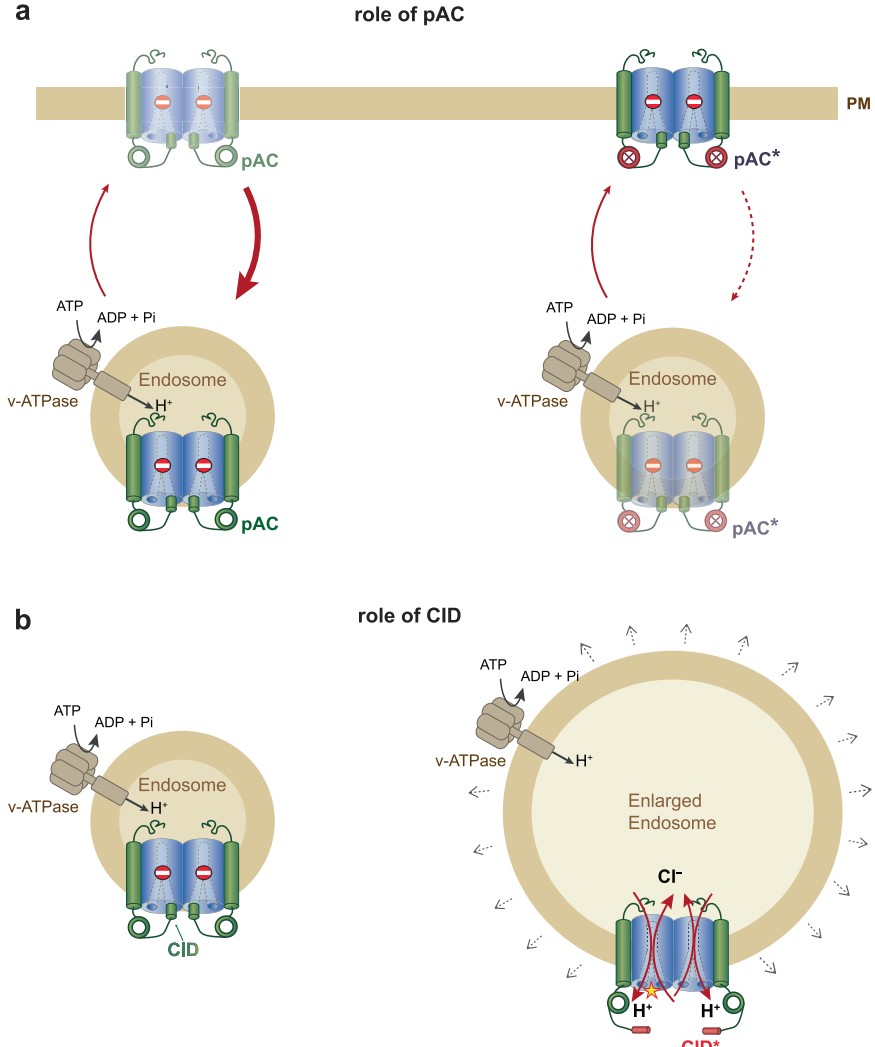

**Fig. 7 | Schematic illustration of selected TMEM9 functions. a** Dimeric ClC-3 to ClC-5 2Cl⁻/H⁺-exchangers associate with T9A or T9B β-subunits. They normally largely inhibit CLC ion transport by blocking the cytoplasmic opening of the Cl⁻ permeation pathway[46] with part of their *Cytoplasmic Inhibitory Domain* (CID) located at the extreme C-terminus. In CLC exchangers, separate pathways for Cl⁻ and H⁺ connect the cytoplasm to the Cl⁻/H⁺ exchange site in the center of the protein[57]. Cytoplasmic *phosphorylated Acidic Clusters* (pAC) of T9 proteins stimulate the removal of vCLC/T9 complexes from the plasma membrane, leading to a predominant endosomal localization. pAC mutations shift the equilibrium to the PM. **b** Disturbing the binding of CID to cytoplasmic parts of vCLCs by either mutating CID or interacting vCLC residues (as with several disease-causing *CLCN* mutations, yellow asterisk) disinhibits (activates) vCLCs. An inside-out proton gradient, generated by the vesicular H⁺-ATPase, causes excessive luminal Cl⁻ accumulation through 2Cl⁻/H⁺-exchange. The resulting osmotic gradient, which inhibits vesicle budding, entails apparent swelling of endolysosomal compartments.

staining pattern of ClC-5 to punctate, endosome-like labeling (Fig. 6c). Co-expressing ClC-5 with T9B^ΔCID or T9A^ΔCID led to massive vacuolization (Fig. 6d), suggesting that vacuole formation by ClC-5 requires its localization to endosomes. By contrast, no vacuolization was observed with ClC-4/T9A^ΔCID (Fig. 6e). ClC-4 needs ClC-3 rather than T9 to traffic to endosomes as heterodimer[29,47]. Co-expressing transport-deficient ClC-3^td with ClC-4 generated giant vesicles which were inhibited by T9A (Fig. 6f). These vacuoles were generated by ion transport through ClC-4 as these are the only transport-competent subunits in ClC-3^td/ClC-4 dimers. Hence not only overactivity of ClC-3, but also of ClC-4 and ClC-5 causes vacuolization when these proteins are directed to endosomal compartments.

These results allowed us to re-examine pathogenic ClC-4 mutants in their native sequence context. We chose ClC-4^R360S (ref. 45) which we had tested with the equivalent ClC-3^R418S mutant (Fig. 5c). T9 co-expression suppressed vacuolization elicited by ClC-3/ClC-4 (Fig. 6g),

but not by ClC-3/ ClC-4^R360S (Fig. 6h), validating our ClC-3 based test of human mutants.

## Discussion

We have identified T9A (TMEM9) and T9B (TMEM9B) as obligatory β-subunits for ClC-3 to ClC-5 2Cl⁻/H⁺-exchangers. They are required for CLC protein stability and regulate diverse trafficking steps and ion transport activity (Fig. 7). Their physiological importance is evident from the lethality of *T9a^−/−/T9b^−/−* mice and from disease-associated *CLCN3* and *CLCN4* mutations that weaken inhibition by T9. Our work uncovered relief from tonic, β-subunit mediated inhibition as an efficient mechanism for regulating endosomal ion transport.

T9 proteins are crucial for endosomal targeting of ClC-3 to ClC-5. We discovered that T9 proteins foster the transport of ClC-5 from the ER to endosomes (Fig. 6c,d). They also prevented vCLC accumulation at the plasma membrane (Figs. 3b, 7), which they reach in a recycling

process[7]. The decrease in surface expression depended on a phosphorylated acidic cluster (pAC) motif. This matches pAC's role in endocytosis[37]. Since pACs regulate diverse trafficking steps[36,37,48], we cannot exclude that other trafficking steps of CLC/T9 complexes, including the ER to endosome transport of ClC-5 (Fig. 6c,d), are influenced by pAC. Of note, the function of pAC motifs depends on phosphorylation of nearby Ser or Thr residues[36,37]. Our work suggests that Tyr phosphorylation may also modulate the function of pACs (Fig. 3f).

The surprising strong inhibition of vCLC ion transport by T9 proteins provides previously unknown possibilities for regulating $Cl^-/H^+$ exchange. The length of the CLC/T9 interface, as deduced here from functional studies and later confirmed by cryo-EM structures[46], suggests that posttranslational modifications may synergistically activate vCLCs. Phosphorylation-mimicking mutations (Fig. 6a,b) indicated that phosphorylation of both vCLC and T9 proteins may activate $Cl^-/H^+$ exchange. The identification of relevant kinases and phosphatases is are important goal for future research.

The strong inhibition of ion transport by T9 proteins is unusual for β-subunits, but makes biological sense. First, it prevents deleterious effects of unchecked endolysosomal $2Cl^-/H^+$-exchange. Second, it allows large increases in transport rates when strong inhibition is relieved by regulatory mechanisms.

Overexpression of ClC-3[38] or of disease-associated activating mutants of ClC-6[11,39] and ClC-7[13,40] generates large lamp1-positive vesicles. Our work extends this observation to ClC-4 and ClC-5, provided they are targeted to endosomes (Fig. 6d, f). Vacuole formation probably involves osmotic swelling by pH- and voltage-driven luminal $Cl^-$ accumulation through electrogenic $2Cl^-/H^+$-exchange. Mechanical stretching of their membrane impairs binding of curvature-recognizing proteins, thereby inhibiting vesicle budding, but not vesicle fusion. Like described for ClC-6[11] and ClC-7[13,40], many giant vesicles were poorly acidified. We propose that they have reached a quiescent state in which a positive luminal potential largely shuts down endolysosomal ion transporters (Supplementary Note).

Given the harmful effects of excessive $Cl^-/H^+$-exchange, it must be strictly controlled. Lysosomal ClC-7 is inhibited by $PI(3,5)P_2$[49]. Disrupting this inhibition entails severe pathology[13,40,49]. With ClC-6, its extreme voltage-dependence[15] may prevent toxic $Cl^-/H^+$-exchange[11,39]. We now found that ClC-3 to ClC-5 use regulated inhibition by associated β-subunits. Although $Cl^-/H^+$ exchange of individual vCLCs is grossly stimulated by the absence of T9, no vacuoles were observed in T9A/B double-KO cells. This may be explained by the concomitant decrease of vCLC protein levels (Fig. 2c). Likewise, it is unclear whether the lethality of $T9a^{-/-}/T9b^{-/-}$ mice is owed to the loss of vCLCs, as suggested by the lethality of vCLC double KO mice[29,50]. The decrease in vCLC expression might be offset, or even overcompensated, by the loss of T9-mediated ion transport inhibition.

The inhibitory effect of the C-terminus involved residues distributed over a stretch of about 20 amino-acids which we named *C*-terminal *I*nhibitory *D*omain (CID). The length of this domain allows, in principle, synergistic effects of posttranslational modifications of several residues. Such modifications probably require low affinity of CLC/CID binding in order for kinases to gain intermittent access. The length of CID agrees well with the positions of human mutants that interfere with inhibition. Single point mutations in either CID or interacting CLC regions only partially destabilized the interaction as revealed by mild functional effects. Effects of point mutations were only detected with the vesicle assay but not with PM currents. This comes down to the sensitivity of these assays. Currents must exceed >10% of maximal ClC-5 currents to be distinguishable from background. By contrast, starting from possibly near-total inhibition of endosomal CLCs, relative increases of ion exchange rates might even span orders of magnitude while lacking

detectable effects on PM currents. Although disinhibition of $Cl^-/H^+$ exchange by human mutations may be small, it suffices to cause severe disease.

It is enthralling to compare CID-mediated regulation of vCLCs to the gating of the ClC-2. The slow hyperpolarization- and swelling-induced opening of this plasma membrane $Cl^-$ channel involves a "ball and chain" mechanism[42]. An inhibitory stretch of its cytoplasmic amino-terminus (the 'ball') interacts with a "receptor" on the cytoplasmic surface of the channel which includes, just like observed here for CID, a portion of the J-K linker[43]. A recent cryo-EM study[44] confirmed these results and revealed that part of the ClC-2 N-terminus blocks the cytosolic opening of the channel's pore. Likewise, our companion cryo-EM study[46] shows that T9A's CID blocks the cytoplasmic mouth of the $Cl^-$ permeation pathway through ClC-3. The blockade of ClC-2 by its N-terminus is clearly reversible because it can be relieved by hyperpolarization and other factors[42,44]. Our data suggest that likewise CID-mediated inhibition of vCLCs is reversible and might be modulated by phosphorylation. The regulation of ion transport of the ClC-2 $Cl^-$ channel and of vCLC $Cl^-/H^+$-exchangers by the N-terminus of the same protein, and C-terminus of associated β-subunits, respectively, constitute an intriguing example of convergent evolution.

Currents reported for ClC-3 to ClC-5 always involved CLC overexpression and thus missed effects of their β-subunits. Our study leads to the reclassification of several disease-associated *CLCN3* and *CLCN4* variants as gain-of-function mutants. These variants caused vacuolization upon overexpression. Activating *CLCN7* mutations can cause vacuolization in patients' tissues[13,40], but it is unclear whether this occurs in *CLCN3* or *CLCN4* neuropathy patients whose mutations we have analyzed here. Excessive endosomal $Cl^-/H^+$-exchange may be pathogenic well below the threshold for giant vacuole formation.

Not only *CLCN3* and *CLCN4* mutations might cause disease by weakening CID-mediated inhibition, but also mutations in *T9* genes. A loss of either T9A or T9B might produce only mild symptoms because both subunits partially substitute for each other. Point mutations affecting CID may cause more harm than a complete loss of one T9 isoform. This is because CLC/T9 association does not require T9 C-termini and bound mutants cannot be replaced by WT counterparts. Since activating mutations in *CLCN3* and *CLCN4* are associated with neurological disease, we expect to find T9 CID mutations in nervous system disorders.

By identifying and characterizing TMEM9 and TMEM9B as β-subunits of ClC-3 to ClC-5, we have uncovered previously unknown mechanisms to regulate trafficking and ion transport of endosomal CLCs (Fig. 7). Crucially, our work has revealed the necessity to tonically suppress endolysosomal $Cl^-/H^+$ exchange activity. Weakening this inhibition can lead to severe cellular and organismal pathology. Several human ClC-3 and ClC-4 mutants underlying neurological disease display reduced sensitivity to T9-mediated inhibition. Our conclusion that the inhibitory C-terminus interacts with various cytoplasmic parts, including residues in CBS domains, the N-terminus, and two intracellular loops of CLCs, is confirmed by cryo-EM structures we have recently obtained[46]. This study also finds that CID-CLC interactions are partially mediated by interposed $PI(3,5)P_2$ molecules[46]. Several aspects of endosomal CLCs must be revisited. Our work creates the basis for understanding the regulation of endosomal $Cl^-$ and $H^+$ transport and its impact on physiology and disease.

## Methods

### High-resolution complexome profiling (csBN-MS)
Plasma membrane-enriched fractions were prepared from two freshly isolated mouse brains, solubilized and processed following the complexome-profiling procedure[51]. Protein complexes (solubilized in ComplexioLyte 47a (Logopharm) with salt replaced by 750 mM aminocapronic acid) were enriched by ultracentrifugation into a sucrose

cushion (20%/50% sucrose), supplemented with 0.125% Coomassie G250 and resolved on a hyperbolic 1-18% preparative native gel (BN-PAGE). The gel lane section of interest was embedded and sliced into 0.3 mm sections using a cryo-microtome (Leica) as detailed in ref. 52. After in-gel tryptic digestion, samples were analyzed on a Q Exactive mass spectrometer coupled to an UltiMate 3000 RSLCnano HPLC system (Thermo Scientific, Germany). Mass spectrometric raw data were processed as in ref. 21 with the following modifications: after off-line correction of precursor mass offsets, MS data was searched with PEAKS 11 (Bioinformatics Solutions, Inc.) against the UniProt mouse reference database (UP000000589). Identification threshold for protein groups was set to 1%, proteins either representing exogenous contaminations (e.g., keratins, trypsin, IgG chains) or identified by less than 5 specific peptides were not further considered. For quantitative evaluation, ion intensities (integrated peak volumes, PVs) were extracted and mass calibrated by MaxQuant v1.6.3 (http://www.maxquant.org) and assigned to peptides after retention time-alignment of runs, consistency-filtered using the Expector procedure[21] and protein abundance profiles determined. Slice numbers were converted to apparent complex molecular weights by fitting of log(MW) of marker complexes versus their observed profile peak maxima slice index using a sigmoidal function. Protein abundance profiles were finally smoothed using a Gaussian filter (width set to 1.4).

## Molecular biology

*Tmem9* and *Tmem9b* cDNAs were cloned by RT-PCR from total RNA isolated from C57BL/6 N mouse kidney using TRIZOL (Invitrogen) and RNeasy columns (Qiagen). RT-PCR used SuperScript™ II reverse transcriptase and oligo(dT)15 primers for cDNA synthesis. PCR products were cloned into pCDNA3.1 (Thermo Fisher). Epitope tag insertions, chimaeras, deletions, and mutations were generated by recombinant PCR or by using the QuikChange kit (Agilent). For site-directed mutagenesis, oligonucleotides were designed using the 'QuickChange Primer Design' tool (https://www.agilent.com/store/primerDesignProgram.jsp) from Agilent. Human ClC-5 cDNA was subcloned into pEGFP-N1 (Clontech), mouse ClC-3c, human ClC-3a, -4, -5, and -7 cDNAs into a modified pEGFP-C1 vector where EGFP was replaced by mVenus. All clones were verified by sequencing the entire ORF.

Three HA-epitopes were fused either to the C-termini of T9A and T9B, or inserted shortly after the predicted cleavable signal peptide (Supplementary Table 1). To detect surface expression of ClC-5, we had previously inserted HA-epitopes into the first extracellular loop of ClC-5[53]. For the present study, this construct is useless because this loop binds the extracellular domain of T9 proteins[46]. We therefore inserted a large flexible 163 residue-long unstructured polypeptide with three HA-epitopes into the extracellular/luminal loop, encoded by sequence generated by DNA synthesis, between helices L and M of ClC-5 (Supplementary Table 1). The mutant was functional, but gave less current than WT ClC-5 and appeared less sensitive to T9 inhibition. The respective constructs were inserted into pTLN[16] and pCDNA for oocyte and HeLa or HEK 293 expression, respectively.

To study which domains of T9A associate to GFP-ClC-5, several chimeras were generated that exchanged different domains (N-, C-, and transmembrane TMD) of T9A with the corresponding domains of mouse Ostm1. Specifically, the T9A N-term encompassed $^1$M-$^{89}$K, the C-term $^{111}$L-$^{183}$S, and the TM from $^{90}$V-$^{110}$M. For Ostm1, N-term ($^1$M-$^{288}$S), C-term ($^{310}$H-$^{338}$T), and TMD ($^{289}$V-$^{309}$L).

## Cell culture and transfection

HEK 293 and HeLa cells were obtained from the Leibniz-Institut DSMZ-Deutsche Sammlung von Mikroorganismen und Zellkulturen GmbH, Germany. Both cell lines were maintained in complete medium (DMEM with 10% FBS and 1% penicillin/streptomycin) at 37 °C and 5% $CO_2$. Cells were transiently transfected using either JetPrime (Polyplus), PEI (polyethylenimine), or Lipofectamine 3000 (Thermo Fisher).

## Generation of monoclonal knockout cell lines

HeLa cell lines with disruptions in *Tmem9* or *Tmem9b* genes were generated using CRISPR/Cas9 using the px330 system[54] and the following guides: 5' GCTGTGCGAGTGCAGGTACG 3'[24] for *T9A*$^{-/-}$ and 5' ATCAGGCCCCCGCACAGGCA 3' for *T9B*$^{-/-}$. Selected clones were analyzed by target site-specific PCR on genomic DNA followed by Sanger-sequencing, revealing frame shifts and early translation terminations. Clones were further confirmed by Western blot and immunofluorescence with custom made antibodies.

## Generation of *Tmem9* and *Tmem9b* knock-out mice

Knock-out *Tmem9* and *Tmem9b* mice were generated using CRISPR-Cas9 genome-editing by the MDC Transgenic Core Facility (Berlin). Briefly, one guide sgRNA 5' GCGGACAGATGCATTTGCAC 3' targeting exon 3 of *Tmem9* and a second guide sgRNA 5' TATGGTGCG-GAAACCTGCTG 3' targeting exon 1 of *Tmem9b* were co-injected by pronuclear microinjection in fertilized C57BL/6 N mouse oocytes that were implanted into foster mothers. One founder carrying a 37 bp deletion in *Tmem9* and one founder with 32 bp deletion in *Tmem9b*, leading to frameshift and early protein translation terminations, were selected. Nucleotide deletions were confirmed by PCR and Sanger sequence analyses. Founders were backcrossed a minimum of three times with C57BL/6 N mice to generate *Tmem9*$^{+/-}$ or *Tmem9b*$^{+/-}$ animals, verified by PCR genotyping and genome sequencing. For PCR genotyping, genomic DNA was amplified in 30 cycles (30 s, 94 °C; 45 s, 60 °C; and 30 s, 72 °C) on a thermal cycler. For *Tmem9a*, primers 5' GCACGTCACTTCATGGATG 3' and 5' CAGCCATTAGCAGTGTCTTAC 3' amplified a 370 bp or 333 bp band for the WT or KO allele, respectively. For *Tmem9b*, primers 5' TAAGGCTTAGGATGCTCTAGAG 3' and 5' AGCCTATGGTGCGGAAACC 3' amplified a 275 bp or 243 bp band for the WT or KO allele, respectively. These were used for the establishment of the single and double KO mouse lines. *Tmem9*$^{-/-}$ and *Tmem9b*$^{-/-}$ mice were viable and fertile. Double KO animals were never found in more than 30 litters, as further confirmed in breeding pairs using homozygous KO mice for one T9 gene and heterozygous for the other.

From double heterozygous (*Tmem9*$^{+/-}$, *Tmem9b*$^{+/-}$) breedings, we obtained a total of 98 embryos at age E12.5 of which 6 were double KO, 6 WT, 4 T9A-KO, and 7 B-KO, in good agreement with the expected ratio (6.25%). The gross morphology of double KO E12.5 embryos was normal. E12.5 embryos were used to generate independent mouse embryonic fibroblasts (MEFs) that gave similar results in our assays.

Absence of protein expression in KO mice was validated by Western blot using custom-made antibodies.

## *Clcn* mouse models

The *Clcn* mouse models used in this study had been generated in the Jentsch lab and were first described in the following publications: *Clcn3*$^{-/-}$ mice, ref. 8; Venus-ClC-3 mice, ref. 29; *Clcn4*$^{-/-}$ mice, ref. 50; and *Clcn5*$^{-/-}$ mice, ref. 31. These mice were maintained in B6/SVJ mixed background.

## Animal experiments

All experiments with mice, including the generation of new mouse lines, breeding, organ collection or perfusion were approved by Berlin authorities (LAGeSo) under licenses G 0111/17-21012, G 0051/22, X 9011/22, T-02-2024, G 0005/19, G 0045/24, O 0380/17 and G 0041/23. Surgery and maintenance of *Xenopus laevis* frogs were approved by LAGeSo under license E 0103/23. Animals were housed in the MDC and FMP animal facilities according to institutional guidelines and under supervision by the authorities.

## Primary culture of mouse bone marrow-derived macrophages (BMDMs) and embryonic fibroblasts (MEFs)

Murine BMDMs were isolated from 6 to 8 weeks C57Bl/6 N adult tibiae and femora from both sexes. Cells were flushed from bone, and erythrocytes lysed in red blood cell lysis buffer (Abcam, ab204733). Remaining cells were resuspended in complete DMEM medium. Four hours after seeding, non-attached cells were cultured in complete DMEM containing 20 ng ml$^{-1}$ recombinant murine M-CSF (Pepro Tech, 315-02). Cells were used 6-14 days later. M-CSF concentration was increased to 100 ng ml$^{-1}$ to induce micropinocytosis and then fixed after incubation for 5-10 min. To obtain embryonic fibroblasts, eviscerated trunks from E11.5-12.5 embryos were minced and incubated in 0.05% trypsin-EDTA (GIBCO) for 30 min at 37 °C and later mechanically homogenized. After trypsin inactivation in complete medium (DMEM plus 10% FBS), cells were cultured up to six passages.

## Evaluation of differentially expressed genes by qRT-PCR

Three 8-10 weeks-old C57Bl/6 N WT and T9A- and T9B-KO littermates from either sex were used per organ. RNA was isolated from tissues, and cDNA synthesized using used SuperScript™ II reverse transcriptase (Thermo Fisher Scientific) and oligo(dT)15 primers. qRT-PCR was run in a StepOne Plus (Applied System Biosystems) (StepOne software v2.3) using the SYBR green PCR master mix (Applied Biosystems). The following pairs of primers were used for qRT-PCR: for *ClCN3* 5′ tgtaactcacaacggacgcctc 3′ and 5′ tattgaagcgggggtcttggttt 3′ (ref. 28), for *ClCN4* 5′ gcgtctcatcgggtttgc 3′and 5′ ttgccacaatgccctcttg 3′ (ref. 8), for *ClCN5* 5′ aatcatcaccaaaaaggatgtgttaa 3′ and 5′ ccatggtccg-caatgtcc 3′ (ref. 55), for *GAPDH* 5′ agcctcgtcccgtagacaaaa 3′ and 5′ tggcaacaatctccactttgg 3′ (ref. 55), for *Tmem9* 5′ gtccgccttacagaaacatca 3′ and 5′ctcgtacctacactcgcagag 3′, PrimerBank-MGH-PGA Harvard U., and for *Tmem9b* 5′ ctatggtgcggaaacctgct 3′ and 5′ gcccaggattctctttataggga 3′, PrimerBank-MGH-PGA, Harvard. Reactions were performed in triplicate with cDNAs from at least three different animal pairs. Fold changes were calculated and geometrically averaged.

## Generation and characterization of Tmem9/9B antibodies

Polyclonal antibodies against different peptides in the C-terminal intracellular domain of mouse TMEM9 (T9A) and TMEM9B (T9B) were generated, both in rabbits and guinea pigs, by Pineda Antibody Service (Berlin) and Eurogentec (Seraing, Belgium). Antigen sequences are depicted in Supplementary Fig. 1a. Peptides were coupled to KLH via N-terminally added cysteines. Sera were affinity-purified against the respective peptides, and their specificity was ascertained by Western blot and immunofluorescence in both WT/KO Hela cells and WT/KO mice.

## Western blot analysis of tissue extracts

Organs were harvested from adult mice of different genotypes and sexes, which were sacrificed by cervical dislocation, and membrane fractions prepared. Briefly, tissues were homogenized in HBS Buffer containing (in mM): 20 Hepes pH 7.4, 150 NaCl, 5 EDTA, 4 Pefabloc (Roth), and Complete® (Roche) protease-inhibitors. After being cleared twice by centrifugation at $900 \times g$ for 10 min at 4 °C, membranes were pelleted at $100,000 \times g$ for 30 min at 4 °C. Pellets were resuspended by sonication in HBS containing 1% Tx-100, incubated 30 min on ice, and then centrifuged for 10 min at $20,800 \times g$ at 4 °C. Protein concentration of the supernatants was determined by BCA, and equal amounts of protein were separated by SDS-PAGE. Equal loading was ascertained by staining for actin or GAPDH.

## Co-immunoprecipitation

HeLa cells transfected with C-terminally HA-tagged T9A and N-terminally tagged Venus-ClC-3a, -3c, --4, -5 or -7 were collected and lysed in HBS buffer containing 1% n-dodecyl-β-maltoside (DDM,

Glycon), plus Pefabloc (Roth) and Complete® protease inhibitor cocktail (Roche). The lysate was cleared by centrifugation at ~72,000 $g$ for 30 min and incubated overnight with Pierce® anti-HA magnetic beads (Thermo Fisher). To study T9A domains able to associate to GFP-ClC-5, HEK 293 cells expressing different HA-tagged chimeras of T9A were processed as above, but lysates were incubated for 2 h with GFP-Trap® Magnetic Agarose beads (ProteinTech). After extensive washing, proteins were eluted with Laemmli buffer and denatured. Lysate equivalent to 10% of input and eluates were loaded and proteins separated by SDS-PAGE and analyzed by Western blotting using antibodies against GFP and HA.

## Multi-epitope affinity purification mass spectrometry (meAP-MS)

Mouse brains from different genotypes (WT, transgenic Venus-ClC-3, $T9A^{-/-}$, and $T9B^{-/-}$) were collected and snap-frozen. Brain tissues were then homogenized at 10 ml/g tissue in buffer containing (in mM): 320 sucrose, 10 Tris-HCl pH 7.5, 1.5 MgCl₂, 1 EGTA, and protease inhibitors, with a Dounce homogenizer and centrifuged for 5 min at 1080 x $g$. Pellets were re-homogenized in 8 ml buffer and centrifuged as before. Combined supernatants were ultracentrifuged (12 min at $150,000 \times g$ (for r(avg), Sorvall S80 AT3). Obtained pellets were hypotonically lysed by resuspension (at 10 ml/g tissue) in buffer containing (in mM): 5 Tris-HCl pH 7.5, 1 EDTA, and protease inhibitors using a syringe with narrow cannula. After incubation for 45 min on ice, lysates were transferred to ultracentrifugation soft tubes, underlayered with 15 ml 0.5 M sucrose and 15 ml 1.3 M sucrose and ultracentrifuged for 45 min at 111,000 $g$ (for r(avg), Beckmann SW32 Ti). Membrane vesicle bands at the sucrose interfaces were collected, diluted 2.5x with 20 mM Tris-HCl pH 7.4, and pelleted by ultracentrifugation (25 min at $150,000 \times g$ (for r(avg), Sorvall Ti 865). Membrane pellets were resuspended in dilution buffer, volume adjusted to 10 mg/ml (Bradford assay, Bio-Rad), and snap-frozen in liquid N₂.

For affinity purification, aliquots of 1-2 mg membrane were solubilized (at 0.9 ml/mg protein) in ComplexioLyte 47 (Logopharm) with protease inhibitors (incubation for 20 min on ice, ultracentrifugation for 12 min at $150,000 \times g$ (for r(avg), Sorvall S80 AT3) and incubated with 10-20 μg of the following antibodies (immobilized on Protein A / Protein G magnetic beads (according to the manufacturer's instructions; Invitrogen): anti-GFP (Roche #11814460001, R&D Systems #AF4240), anti-TMEM9 "T9AC2" rb 1, anti-TMEM9 "T9AC2" rb 2, anti-TMEM9 "T9AC2" gp 1, anti-TMEM9 "T9Act"-rb 2, anti-TMEM9B (Proteintech #24331-1-AP), anti-TMEM9B "T9BC2", anti-TMEM9B "T9BC1", anti-TMEM9B "T9BCt"-gp2 and different mixtures from normal rabbit / mouse / goat IgG (Upstate and Santa Cruz) as negative controls. After incubation for 3 h on a rotating wheel, beads were transferred to new reaction tubes, washed twice 10 min with 1 ml ComplexioLyte 47 dilution buffer (Logopharm), and eluted twice with non-reducing Laemmli buffer (100 mM DTT added after elution). Samples were shortly run on SDS-PAGE gels, silver-stained, and higher- ( > 50 kDa) and lower molecular weight ( < 50 kDa) sections were excised for in-gel tryptic digestion (sequencing grade trypsin, Promega).

Peptide samples were dissolved in 0.5% trifluoroacetic acid and analyzed with an Orbitrap Elite mass spectrometer equipped with an autosampler Ultimate 3000 nano-HPLC (all Thermo Fisher Scientific; C18 pre-column (PepMap100, 5 mm particles), SilicaTip emitter (75 mm i.d., 8 mm tip, New Objective Inc) manually packed with ReproSil-Pur 120 ODS-3 (C18; particle size 3 μm; Dr. Maisch HPLC, Germany). Bound peptides were eluted with an aqueous-organic gradient (from 0.5% acetic acid to 80% acetonitrile / 0.5% acetic acid) and directly electrosprayed (2.3 kV, positive ion mode) into the mass spectrometer. Specific settings were: full precursor scan m/z = 370-

1700, target value 106 ions, nominal resolution 240,000; up to 10 CID MS/MS fragment spectra/cycle (target value 2000), dynamic exclusion: 30 s, mono-isotopic precursor selection enabled.

MS data was processed as described above and searched against the UniProt mouse reference database (UP000000589) using PEAKS 11 (Bioinformatics Solutions, Inc.). FDR threshold for identified protein groups was set to 1%; exogenous contaminations (e.g., keratins, trypsin, BSA, IgGs) or proteins identified with less than three specific peptides were not further considered. Label-free quantification of proteins was done as described above; protein abundance was calculated as abundance norm values[21]. To identify target-specific interaction partners, two parameters were calculated from these abundance values for each protein: (i) enrichment in WT versus IgG and target knockout control APs relative to enrichment of the respective AP target protein (target-normalized ratio, tnR[22] and abundance in IgG and target KO control APs relative to the abundance of the respective AP target protein in WT (target-normalized abundance, tnA)). All pairs of AP versus control (4 ×3 for TMEM9B and 4 ×4 for TMEM9A) were evaluated collectively in t-SNE (for dimensionality reduction) plots using consistency as an additional, stringent criterion using the BELKI software suite (https://github.com/phys2/belki).

For determination of phosphorylation sites, Mascot (Server 2.6.2, Matrix Science Ltd) and PEAKS 11 database search results from TMEM9, TMEM9B, and Venus-ClC3 AP-MS datasets were evaluated. Phosphosites were assigned to these target protein sequences only when at least two specific and high-quality phospho-peptide spectra were verified by manual inspection.

## Immunocytochemistry

Cells were seeded on glass coverslips and transfected with PEI or Jet-PRIME® (Polyplus). 24–48 h after transfection, cells were fixed with methanol or 1% PFA in phosphate-buffered saline (PBS) for 10 min, permeabilized with 0.1% saponin and blocked in 3% goat serum (NGS), 2% BSA in PBS and incubated overnight at 4 °C with primary antibodies (See Supplementary Table 2) in 0.05% saponin in 3% BSA in PBS. Subsequently, cells were incubated for 1 h at room temperature (RT) with secondary antibodies coupled to different fluorophores and nuclei stained with DAPI, extensively washed, and finally mounted on slides. Images were acquired with a confocal microscope with a 63× NA 1.4 oil-immersion lens (LSM880, Zeiss).

## Immunohistochemistry of kidney sections

After deep anesthesia, WT, *T9a*$^{-/-}$, *T9b*$^{-/-}$, and *Clcn5*$^{+/-}$ mice[31] were perfused for 3 min with PBS followed by 3 min with 1% paraformaldehyde (PFA) in PBS through the left chamber of the heart. Dissected kidneys were post-fixed with 1% PFA overnight and shock-frozen using isopentane on dry ice. 4 μm-thick kidney cryosections were cut using the CryoStarNX70 (Thermo Fisher) and mounted on microscope slides. Sections were blocked for 2 h with 5% NGS, 0.25% Triton X-100 in PBS at RT, and then incubated overnight at 4 °C with primary antibodies (See Supplementary Table 2) in 1% BSA in PBS. After three 10-minute washes with PBS containing 0.25% Triton X-100, sections were incubated with secondary antibodies and DAPI in 3% BSA for two hours at RT, washed, and mounted with Fluoromount-G. Images were acquired with the confocal LSM880 microscope.

## Live imaging with lysotracker and lysosensor yellow/blue

Hela WT, *T9A*, *T9B-* or *T9A/T9B*-KO cells or mouse primary embryonic WT and *T9A/T9B*-KO fibroblasts (MEFs) were seeded on glass-bottom (1.5H) 35-mm live imaging dishes (MatTek). After 18–24 h, cells were incubated with 75 nM Lysotracker Red DND-99 (Invitrogen, L7528) in complete DMEM medium for 2 h at 37 °C. Nuclei were stained for 5 min with Hoechst, then washed twice with PBS and immediately imaged at 37 °C at the confocal microscope in prewarmed HBSS buffer. As

control, 100 nM Bafilomycin (Sigma) was included in some samples to alkalinize lysosomes.

For ClC-3a-induced vacuolization analysis using incubation with Lysosensor®Yellow/Blue DND-160 (Invitrogen, #L7545), non-transfected HeLa cells, or co-transfected with Venus-ClC-3a (WT or mutants) plus T9s (WT or mutants), or CD4 as control, were used. Venus was replaced by mCherry in ClC-3 plasmids when co-expressed with Venus ClC-4 WT or R360S mutant. For confocal imaging, cells were seeded in glass-bottom MatTek imaging dishes. After 48 h transfection, cells were incubated for 15 min at 37 °C with 5 μM Lysosensor in complete DMEM medium, then washed twice with PBS and immediately imaged in prewarmed HBSS medium. Cells were imaged with the LSM880 confocal microscope using $\lambda_{ex} = 405$ nm and $\lambda_{em} = 511\text{-}588$ nm for Lysosensor, and $\lambda_{ex} = 488$ nm for Venus.

## Detection of large vacuoles by microscopy

To define enlarged vacuoles, we set a minimum size threshold (cross-sectional area >1.0 μm² in confocal microscopy). Immunocytochemistry revealed that enlarged vacuoles consistently co-stained with Lamp-2. Each mutant was first examined by live cell imaging using Venus-ClC-3 and Lysosensor fluorescence in a minimum of two independently transfected coverslips. The presence of cells displaying large vesicles was further confirmed in at least three independent experiments of fixed transfected cells in immunocytochemistry using anti-GFP antibodies (to detect Venus-ClC-3) and anti-T9 antibodies (to confirm the co-expression of the T9A or T9B on these vacuoles). Importantly, cells displaying large vacuoles were never observed with 1:1 co-transfection of Venus-GFP and WT T9A or T9B.

The percentage of cells displaying >5 large vacuoles was low ($\approx 20\text{-}25\%$) on any given coverslip transfected with Venus-ClC-3. The percentage of vacuole-positive cells was lower when partially inhibition-resistant variants of Venus-ClC-3 and T9A or T9B were co-expressed. There appeared to be a rough, positive correlation between the size of vacuoles in individual cells and the number of cells having large vacuoles, which can be taken as rough measure for "efficiency of disinhibition". For instance, when co-transfected with T9A or T9B, the JK-linker substitution mutant ClC-3$^{JK-C2}$ led to larger vacuoles, and to a larger number of vacuole-positive cells, than the ClC-3 mutants R418S and even more so than K422G, indicating a lower resistance to inhibition of these single point mutants. This observation correlates with the more quantitative FC analysis: with ClC-3$^{JK-C2}$ a population of Lysosensor-positive (vacuole-positive) cells was easily detected (Fig. 5a) (which was also the case with CID alanine scanning (Supplementary Fig. 8c). For single point mutants, cells having marked Lysosensor® fluorescence were hidden in the background of the few cells with high fluorescence that was also observed in non-transfected cells. To detect the low degree of disinhibition observed with human pathogenic mutations, we had to resort exclusively to microscopical analysis. A minimum of 6 images per preparation were acquired and analyzed by ImageJ (FIJI).

## Flow cytometry analysis

For flow cytometry (FC), cells were seeded in 6 cm dishes and were transfected as above with Lipofectamine 3000 (Invitrogen, #L3000015). After 48 h, cells were harvested with detachment buffer (20 mM EDTA in Ringer's solution), washed by centrifugation ($300 \times g$, 2 min, at RT), and incubated with 5 μM Lysosensor® for 15 min at 37 °C while in suspension in isotonic Ringer's solution. After washing in Ringer's solution, cells were resuspended in 5% BSA in Ringer's solution and analyzed by FC using an LSRFortessa (BD Biosciences).

Data were acquired from a minimum of 50,000 events, and processed by gating as shown in Supplementary. Fig. 8a. GFP expression was detected using a 488 nm laser and 530/30 filter, and Lysosensor fluorescence was measured with a 405 nm laser and 525/50 filter. FlowJo™ software (v10.6.1) was used for data analysis, and fluorescent

intensity values were used for quantification. Lysosensor fluorescence was assessed in GFP-(Venus) positive cells. Results are shown as intensity histograms of individual experiments, and mean values of the percentage of 'high Lysosensor intensity' cells from several independent experiments (insets). To obtain this percentage, a bisector tool was used to set a threshold defined as the mean fluorescent intensity (MFI) plus two standard deviations (mean + 2 SD). This threshold was determined for the condition in which large vesicles were suppressed, i.e., ClC-3 plus inhibitory T9 protein, or ClC-3 *unc* or *td* mutants, and then applied to all conditions in that experimental batch. Data were analyzed in GraphPad Prism (v10.3.1). Nonparametric statistical tests were applied. The Mann-Whitney U test was used to compare the percentage of high Lysosensor intensity cells between control and experimental conditions. Statistical significance was set at $p < 0.05$. Data are presented as scattered dot plots of mean with SD, and all comparisons were performed using two-tailed tests.

### Ratiometric measurement of lysosomal pH

Hela WT or $T9A^{-/-}$ or $T9A^{-/-}/T9B^{-/-}$ cells seeded on glass-bottom (1.5H) imaging dishes (MatTek) were loaded overnight with 0.5 mg/ml Oregon Green 488 coupled 10 kDa dextran (Cat. D7170, Thermo Fisher Scientific) in complete medium (DMEM plus 10% FBS) at 37 °C, and then chased for 2 h. After washing, cells were imaged in imaging buffer containing (in mM): 135 NaCl, 5 KCl, 1 CaCl$_2$, 1 MgSO$_4$, 10 D-glucose, and 25 Hepes pH 7.4 at RT. Images were acquired at $\lambda = 488$ and 435 nm. In situ pH calibration was performed in each sample dish by incubating for 5 min at RT with different pH solutions in the presence of 10 μM Nigericin (Merck) and 10 μM Monensin (Merck). Images were acquired using a Nikon Ti Eclipse widefield epifluorescence-TIRF microscope, Prime 95B camera (Teledyne Photometrics), and a PE4000 CoolLed fluorescence lamp with two excitation filters: 436_20 (Zeiss 0440586) and 480_40 (Zeiss 0440583). Images were analyzed using a ComDet v0.5.5 plugin in FIJI Image J.

### Surface expression in *Xenopus* oocytes

A previously described luminescence assay[53,56] was used to detect the surface expression of proteins carrying extracellular HA-tags. Defolliculated oocytes from *Xenopus* frogs were injected (50 nl/oocyte) with cRNA transcribed from pTLN vectors[16] using mMessage mMachine kit (Thermo Fisher Scientific). Generally, they were co-injected with 10 ng of ClC-5 and 10 ng of T9s or the non-interacting type I membrane protein CD4 as negative control. To compare the effects of different mutants, oocytes from the same batch were injected with the respective combinations on the same day.

Surface expression was measured 2–3 days after injection. Oocytes were placed for 30 min in ND96 with 1% BSA at 4 °C to block unspecific binding, incubated for 60 min at 4 °C with 1 μg/ml rat monoclonal anti-HA antibody (3F10, Roche, in 1% BSA/ND96), washed at 4 °C, and incubated with horseradish peroxidase-coupled secondary antibody (goat anti-rat Fab fragments, Jackson ImmunoResearch, in 1% BSA for 30–60 min at 4 °C). Oocytes were washed thoroughly (1% BSA, 4 °C, 60 min) and transferred to ND96 solution without BSA. Individual oocytes were placed in 50 μl of Pierce SuperSignal™ ELISA Femto Substrate solution (Thermo Fisher Scientific) and incubated at RT for 1 min. Chemiluminescence was quantified in a Turner TD-20/20 luminometer (Turner Design).

### Western blot analysis of oocyte extracts

Oocytes previously used for surface expression measurements were pooled and stored at −20 °C. For Western blot analysis, thawed oocytes were homogenized in ice-cold homogenization buffer containing (in mM) 250 sucrose, 0.5 EDTA, 5 Tris-HCl (pH 7.4), and protease inhibitor mix (Complete®, Roche Molecular Biochemicals; and Pefabloc®, Roth). Egg yolks were removed by three consecutive centrifugation steps at 1,000 × g for 2 min each. Protein concentrations

were determined using the BCA assay, and equal amounts of protein were analyzed by SDS-PAGE and Western blotting. Protein loading was verified by Ponceau staining.

Expression levels of HA-tagged proteins were determined using the monoclonal 3F10 antibody (Roche) in Western blots that were quantified using ImageJ/Fiji. Lane intensity profiles were generated and band intensities measured. Protein expression was quantified as area under the curve (AUC) in arbitrary units and normalized to total AUC.

Surface expression of HA-ClC-5 or HA-T9B was normalized to total protein expression of the respective oocyte samples. Statistical analysis used the Mann–Whitney test with false discovery rate (FDR) correction using the Benjamini–Hochberg procedure.

### Two-electrode voltage clamp in *Xenopus* oocytes

Plasma membrane currents were measured in *Xenopus* oocytes 2–3 days after injection using standard two-electrode voltage clamp. Recording pipettes were filled with 3 M KCl. Currents were recorded using a Turbo Tec-03 X voltage clamp amplifier (npi electronics) and pClamp10 software (Molecular Devices).

Currents were individually background-subtracted. This proved particularly important for the low currents obtained by HA-ClC-5. Since ClC-5 yields virtually no current between −100 and −20 mV (*32*), currents in this range represent background currents. These were fitted by linear regression individually for each oocyte and a corresponding ohmic current was subtracted individually from each current recording. Water-injected control oocytes ascertained that background currents were ohmic until +80 mV. The few recordings in which tail currents indicated the activation of oocyte Ca$^{2+}$-activated Cl$^-$ currents at positive potentials were discarded.

### Patch clamp recordings in mammalian cells

HeLa $T9A^{-/-}/T9B^{-/-}$ cells on glass coverslips were transfected with Lipofectamine™ 3000 (Thermo Fisher Scientific). GFP-ClC-5 (in pEGFP-N1) and non-tagged T9 proteins (in pcDNA3.1) were used for transfection.

Currents were recorded 24 h after transfection in whole-cell configuration at room temperature using a MultiClamp 700B patch-clamp amplifier/Digidata 1550B digitizer and pClamp 10 software (Molecular Devices). GFP-positive cells were selected for recording. Patch pipette solution was (in mM): 140 CsCl, 5 EGTA, 1 MgCl$_2$, 10 HEPES (pH adjusted to 7.2 with CsOH). The extracellular solution contained (in mM): 150 NaCl, 6 KCl, 1 MgCl$_2$, 1.5 CaCl$_2$, 10 glucose. Series resistance was compensated by 60%. During acquisition, recordings were filtered with a low pass Bessel filter at 4 kHz and sampled at 10 kHz. Voltage step protocol consisted of 0.5 s voltage steps starting from −100 to +140 mV in 20-mV increments from a holding potential of −30 mV. Data were analyzed using the Pandas 2.2.2 and SciPy 1.13.1 libraries for Python 3.12 programming language (Python Software Foundation).

### Data analysis and statistics

For electrophysiology and surface expression, data were analyzed using the Pandas 2.2.2 and SciPy 1.13.1 libraries for Python 3.12 programming language (Python Software Foundation). Statistical analysis was performed using non-parametric Mann–Whitney test with false discovery rate corrected using the Benjamini–Hochberg procedure.

### Ethics statement

All animal experiments reported in this work have been performed in accordance to German law, and corresponding permits have been obtained by Landesamt für Gesundheit und Soziales Berlin (LAGeSo).

### Reporting summary

Further information on research design is available in the Nature Portfolio Reporting Summary linked to this article.

## Data availability

All relevant data supporting the findings of this study are available within this article, its supplementary information, and Source Data which are provided with this paper. The mass spectrometry proteomics data have been deposited to the ProteomeXchange Consortium via the PRIDE partner repository with the dataset identifier PXD061285 and 10.6019/PXD061285. Requests for materials should be sent to TJJ. Source data are provided with this paper.

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

## Acknowledgements

We thank Carolin Backhaus and Juliane Metzke for technical assistance, Jennifer Lück, Mariia Zeziulia and Karen López-Cayuqueo for initial experiments with TMEM9 and TMEM9B, Maya Polovitskaya for help with electrophysiology, evaluation of results and Fig. 7 preparation, and Fabian Thöne and Martin Lehmann for help with FC analysis. This work was funded in part by the European Research Council (ERC) Advanced Grant 740537 (VolSignal) and the Deutsche Forschungsgemeinschaft (DFG) JE164/12-2 and under Germany's Excellence Strategy–EXC-2049 Project ID 390688087 (NeuroCure), and the Fondation Louis-Jeantet to T.J.J. Work by R.K.H. is supported by NIGMS R01-GM141553 and NIH-National Cancer Institute Cancer Center Support Grant P30-CA008748. B.F. is supported by grants of the DFG (FA 332/15-1, 16-1 and 21-1), and the CRC/TRRs 1453 and 152. M.S. was funded by Walter Benjamin Programme of the DFG.

## Author contributions

Conceptualization, R.P-C, V.V., S.K., M.S., R.K.H., B.F., and T.J.J.; Methodology R.P-C., V.V., S.K., F.W.S., U.S., B.F., and T.J.J.; Formal Analysis, R.P-C., V.V., S.K., F.W.S., U.S., B.F., and T.J.J.; Investigation, R.P-C., V.V., S.K., F.W.S., U.S., B.F., T.J.J.; Writing—Original Draft, T.J.J., R.P-C, S.K.; Funding Acquisition, M.S., B.F., R.H. and T.J.J.

## Funding

## Competing interests

R.H. is a consultant for F. Hoffmann-La Roche Ltd. The remaining authors declare no competing interests.
