## [Transparent Peer Review file · Nature Communications]

Endosomal chloride/proton exchangers need inhibitory TMEM9 β -subunits for regulation and prevention of disease-causing overactivity

Corresponding Author: Professor Thomas Jentsch

Version 0:

Reviewer comments:

Reviewer #1

(Remarks to the Author)

The CICs family of 2Cl⁻/H⁺ exchangers transport Cl⁻ and H⁺, mostly in intracellular organelles including endosomes and lysosomes, and play fundamental roles in cell biology. Mutations in them are associated with many diseases. Several CICs including CIC7 have been known to also include functionally important auxiliary subunits, but the subunit compositions of other CICs such as CIC3 and CIC5 have been elusive. In this paper, the authors report a set of comprehensive and well performed studies that identify two proteins TMEM9A (T9A) and TMEM9B (T9B) as novel “auxiliary” subunits of the CICs. The studies also reveal some of the mechanisms by which T9s modulate CIC functions, and demonstrate the vital function of T9s in the survival of the animals using knockouts. In addition, the authors found that some of the disease-causing CIC mutations in humans affect the CIC-T9 interaction, providing a mechanistic explanation for the diseases.

The evidences the authors provided to demonstrate that the T9 proteins are cardinal subunits of the CICs are remarkably strong, and include similar expression/localization patterns, protein interaction, mutual dependence of protein levels and trafficking, strong mouse phenotypes when knocked out and, more importantly, very striking functional influence of T9s on CIC activity. The experiments are well performed and well presented. The studies represent major milestones in the studies of CIC transporters and are of major interest to cell biologists and neurobiologists interested in endosome/lysosome-related diseases such as neurodegeneration.

I only have several minor comments:

- 1) It'd be useful to further clarify how T9Bs inhibit CIC function. On the plasma membrane (PM), it's clear that T9Bs can “indirectly” decrease CIC function by decreasing CIC localization. The major evidence supporting that T9Bs also “directly” inhibit CIC activity is from the finding that the Δ Cint_{prox} T9 mutant (and perhaps also Δ pAC), unlike the wild-type T9s and most of the other T9 mutants, increases the fraction of PM CIC-5 protein, yet drastically decreases the associated currents (Fig. 2d & h). However, these results can have other explanations. Have the authors tested whether expression of the C-terminus alone of T9s (even better with fusion protein in the patch pipette if possible) is sufficient to suppress CIC currents without affecting CIC localization?
- 2) Fig. 1c, to make it easier for the readers, it'd useful to include a schematic diagram to illustrate the Ostm1-T9 chimeras. In addition, it appears that even Ostm1 CoIP'ed some GFP-CIC5; is this non-specific?
- 3) Fig. 1f, in the double KO MEF cells, there still seems to be considerable amount of signals generated with the T9A antibody (top panel), different from that observed in kidney (Fig. 1e). Is this an antibody specificity issue?
- 4) Fig. 1j, it'd be useful to include T9 KO tissues as a control for the antibody immunostaining specificity, in particular that the authors' data disagrees with previous report of lysosomal localization of the protein.
- 5) Fig. 2b, d, e, f, g, it'd be useful to also include the non-normalized, total amount of protein, perhaps as supplemental data. This information can be used to interpret whether T9s and the mutants also affect the total amount of proteins, in addition to

distribution, particularly given that the protein levels of CICs are affected by T9s in the animals.

6) Fig. 2e upper, the Y-axis would be more clear when labeled as “(norm. to CIC-5HA-T9B WT)”.

7) Fig. 2h, the “rescue” of the CIC currents by T9 C-terminal tagging was used to conclude the “inhibitory” effect of the C-terminus of T9. Does the HA tagging also affect the PM expression level of CICs?

8) Fig. 3, compared to the effects of T9s on CICs on the PM where direct electrophysiological recordings were used, those on the endosomes, the native environment of some of the CICs, were less quantified by using vacuole generation instead of patch clamp recording. To minimize the uncertainty, the authors perhaps should also include data quantifying the amount of CIC proteins on the organelle with and without T9s.

9) it'd be useful to show data to quantify the embryonic lethality by simply including the number of double homozygous embryos found in E12.5

10) Methods/Knockout generation: how many independent founders were used and how many, if any, generations of backcrossing were performed before experimental use?

Reviewer #2

(Remarks to the Author)

In this highly interesting manuscript, the authors have characterized the proteins TMEM9A and B (T9A/B) as β -subunits of the vesicular CLC transporters (vCLCs) CIC3, 4 and 5 and defined their role in trafficking and the regulation of transport. In a comprehensive set of conclusive experiments, they have initially identified T9A/B as co-migrating proteins with vCLCs in brain extracts by complexome profiling and multi-epitope affinity purification. They have also shown that mice containing a knockout of either T9A or B are viable while a double knockout leads to embryonic mortality, suggesting that these proteins are essential but can to some extent replace each other. In immunocytochemistry experiments it was shown that T9A/B target vCLCs to endosomes and that this targeting requires an acidic cluster at the C-terminus. Besides its role in trafficking the authors also showed that T9A/B inhibits the transport function vCLCs via an interaction with their C-terminus. Finally, previously described disease-causing mutations of vCLCs were found to interfere with the interaction with T9A/B. The study is accompanied by a complementary manuscript which provides a structural explanation for the described properties. Although the role of T9B as regulator of CIC3 and 4 was recently described, this work provides comprehensive functional and cell biological insight that goes far beyond the previous study. I thus consider this work as strong candidate for publication in Nature Communication with only minor modifications.

I have the following (minor) remarks the authors should address:

- Fig. 1b, It is not clear what the axes refer to.
- Figure 1 is extensive and could be split into two figures.
- Line 85: Renal expression of CIC5 is dependent on T9B
- What is meant with CIC-4 needs CIC-3 for stability? (line 92)
- Line 106: Please state explicitly for which compartments rab2 and lamp2 serve as markers. What is the conclusion of the poor colocalization of TMEM9A/B with both markers?
- The conclusion stated in line 234-236 is not immediately clear and could be better explained
- It would be helpful if the authors refer to figures in the discussion to clarify their arguments.
- Although the role of the TM helix of T9A/B in the interaction with vCLCs was demonstrated here, the length of the interaction region referred to in line 300-301 was shown conclusively in the accompanying manuscript.
- It would be helpful if the authors would include a discussion figure that schematically summarizes the conclusions of the work.

Version 1:

Reviewer comments:

Reviewer #1

(Remarks to the Author)

The authors have satisfactorily addressed my previous comments and, in my opinion, those of the other reviewer. The manuscript remains to be very strong and is appropriate for publication.

Reviewer #2

(Remarks to the Author)

The authors have addressed all remarks in a satisfactory manner. The manuscript has improved and can be accepted for publication.

Point-by-point responses to reviewers (Planells-Cases et al.)

Both reviewers were very positive and had only minor comments for improving the manuscript.

Reviewer #1 (Remarks to the Author)

The CICs family of 2Cl⁻/H⁺ exchangers transport Cl⁻ and H⁺, mostly in intracellular organelles including endosomes and lysosomes, and play fundamental roles in cell biology. Mutations in them are associated with many diseases. Several CICs including CIC7 have been known to also include functionally important auxiliary subunits, but the subunit compositions of other CICs such as CIC3 and CIC5 have been elusive. In this paper, the authors report a set of comprehensive and well performed studies that identify two proteins TMEM9A (T9A) and TMEM9B (T9B) as novel “auxiliary” subunits of the CICs. The studies also reveal some of the mechanisms by which T9s modulate CIC functions, and demonstrate the vital function of T9s in the survival of the animals using knockouts. In addition, the authors found that some of the disease-causing CIC mutations in humans affect the CIC-T9 interaction, providing a mechanistic explanation for the diseases.

The evidences the authors provided to demonstrate that the T9 proteins are cardinal subunits of the CICs are remarkably strong, and include similar expression/localization patterns, protein interaction, mutual dependence of protein levels and trafficking, strong mouse phenotypes when knocked out and, more importantly, very striking functional influence of T9s on CIC activity. The experiments are well performed and well presented. The studies represent major milestones in the studies of CIC transporters and are of major interest to cell biologists and neurobiologists interested in endosome/lysosome-related diseases such as neurodegeneration.

We thank the reviewer for appreciating the novelty and impact of our manuscript.

I only have several minor comments:

1) It'd be useful to further clarify how T9Bs inhibit CIC function. On the plasma membrane (PM), it's clear that T9Bs can “indirectly” decrease CIC function by decreasing CIC localization. The major evidence supporting that T9Bs also “directly” inhibit CIC activity is from the finding that the Δ Cint_{prox} T9 mutant (and perhaps also Δ pAC), unlike the wild-type T9s and most of the other T9 mutants, increases the fraction of PM CIC-5 protein, yet drastically decreases the associated currents (Fig. 2d & h). However, these results can have other explanations. Have the authors tested whether expression of the C-terminus alone of T9s (even better with fusion protein in the patch pipette if possible) is sufficient to suppress CIC currents without affecting CIC localization?

Thank you for this proposal. We indeed considered performing such experiments, similar to the elegant experiments by Zagotta and Aldrich to nail down the ‘ball-and-chain’ mechanism for N-type inactivation of the shaker potassium channel (DOI: 10.1126/science.2122520). However, our experiments indicated that the interaction between T9 C-termini and CLCs may not be strong enough to observe such an effect (the local concentration of T9 C-termini is much higher when T9 proteins are bound via their TMDs to the CLC than possible with the presence of isolated C-termini or a corresponding peptide in the cytosol).

Moreover, in unpublished experiments, we had previously tried this approach with the N-terminal peptide of ClC-2 which acts via a similar mechanism on ClC-2 ion transport (DOI: 10.1093/emboj/16.7.1582 ; DOI: 10.7554/eLife.90648), but could not observe an inhibition.

Of note, Zagotta & Aldrich used inside-out patches for their recordings, which allows rapid exchange of solutions containing very high concentrations of peptides with control solutions. To the best of our knowledge, nobody has ever been able to observe ClC-5 currents in excised patches. This is not unexpected since the unitary conductance of electrogenic exchangers such as ClC-5 is much lower than those of e.g. shaker K⁺ channels. Based on these arguments, we opted for not trying this approach.

Many experiments in our paper, including the effects of alanine scanning and deletions of T9 C-termini on PM currents of co-expressed ClC-5 (Fig. 3c, i) and the effect of phosphomimicking mutations of C-terminal T9B residue S190 (Fig. 6b), leave little doubt that T9 C-termini directly inhibit ClC ion exchange. Crucially, the direct inhibition of Cl permeation by the C-terminus of T9 proteins is beautifully confirmed by the cryo-EM structures described in the companion paper by Schrecker et al. (NSMB, in revision). That paper also confirms our functional identification of interacting residues described in the present paper. The cryo-EM structures, which we obtained in the final stage of preparing our manuscript, leave virtually no doubt that our interpretation is correct.

2) Fig. 1c, to make it easier for the readers, it'd useful to include a schematic diagram to illustrate the Ostm1-T9 chimeras.

Thank you for this proposal. We have now added such diagrams to the revised Fig. 1e. Since we split the original Fig.1 into Fig. 1 and Fig. 2, as suggested by reviewer 2, we have now enough space to add such schemes.

In addition, it appears that even Ostm1 CoIP'ed some GFP-ClC5; is this non-specific?

We believe that the reviewer mistakenly interpreted the GFP-lane in Fig. 1c (now Fig 1e) as indication for Ostm1 precipitating ClC-5. The GFP blot represents detection of GFP-ClC-5, which was directly precipitated. The proteins Co-IPed by GFP antibodies are shown below, probed with HA antibodies. All T9A-Ostm1 chimeras, as well as T9A and Ostm1, carried an HA epitope at the C-terminus to allow similar detection efficiencies. Thus, Ostm1 did not co-precipitate with ClC-5. For clarity, we have now inserted in brackets in the respective figure panels which blot corresponds to immunoprecipitated "(IP-ed)" or to co-immunoprecipitated "(Co-IPed)" proteins.

3) Fig. 1f, in the double KO MEF cells, there still seems to be considerable amount of signals generated with the T9A antibody (top panel), different from that observed in kidney (Fig. 1e). Is this an antibody specificity issue?

The reviewer is right, the bands detected with our T9A antibody in Fig 1f (now Fig. 2c) the double KO MEFs must be unspecific. It cannot be a spill-over from the WT lane next to it because the intensity ratios of the upper and lower bands are different. The KO cells were confirmed by genomic sequencing of the T9 loci. Furthermore, we established several independent MEF lines from different embryos (confirmed as KO by

sequencing) and observed this band in all independent isolates, see Fig. R1 for the reviewer. These data are also shown in the uncropped western blots in the data supplement file.

Fig. R1 for the reviewer: Western blots for CIC-3 (top) and T9A (bottom) comparing protein extracts from 4 different WT and *Tmem9a*^{-/-}/*Tmem9b*^{-/-} double-KO embryos. Note that the non-specific band detected with the T9A antibody is present in all four independent isolates. The lanes in the red box are the ones displayed in the main figure. This demonstrates that the band mentioned by the reviewer is caused by antibody cross-reactivity rather than a contamination with WT MEFs.

4) Fig. 1j, it'd be useful to include T9 KO tissues as a control for the antibody immunostaining specificity, in particular that the authors' data disagrees with previous report of lysosomal localization of the protein.

As suggested, we now added to the previous Fig. 1j, now Fig. 2g, the new panel Fig. 2h. It shows images of proximal tubules from T9a^{-/-} and T9b^{-/-} mice, each stained with antibodies against T9a and T9b, demonstrating the specificity of the antibodies.

5) Fig. 2b, d, e, f, g, it'd be useful to also include the non-normalized, total amount of protein, perhaps as supplemental data. This information can be used to interpret whether T9s and the mutants also affect the total amount of proteins, in addition to distribution, particularly given that the protein levels of CICs are affected by T9s in the animals.

We agree that it is useful also to show the total amount of proteins expressed in the oocyte experiments, and not just the surface expression. To that end, we had kept (frozen) the oocytes that had been used in our experiments. The results of the Western blots, as well as surface expression normalized to protein levels, are now displayed in the new Suppl. Fig. 5 The normalization to total protein expression does not change the conclusions of our paper. For PM currents of CIC-5, normalization to surface expression. rather than to total protein levels. is preferred to yield a measure of current per transporter.

6) Fig. 2e upper, the Y-axis would be more clear when labeled as "(norm. to CIC-5HA-T9B WT)".

Thank you for the proposal. We have changed the Y-axis to CIC-5 + HA-T9B(WT) in what is now Fig 3e.

7) Fig. 2h, the “rescue” of the CLC currents by T9 C-terminal tagging was used to conclude the “inhibitory” effect of the C-terminus of T9. Does the HA tagging also affect the PM expression level of CLCs?

First, we would like to state that the ‘rescue’ of CLC-5 currents by T9-HA, compared to T9 WT, was only the initial indication that T9 C-termini are inhibiting CLC transport. This observation triggered our mutagenesis study of the effect of the C-terminus, followed by the elucidation of the interaction of T9 C-termini with CLC-3 by cryo-EM (this required the removal of C-terminal amino-acids from the initial T9 construct which resulted from a C-terminal tag used for protein purification; with that first construct, the C-terminus could not be resolved in the CLC-3/T9A complex).

Following the suggestion of the reviewer, we now tested the effect of the C-terminal HA-tag (added to T9) on the surface expression of co-expressed CLC-5. Since the HA-tag probably interferes with the function of the CID domain, we additionally studied the Δ CID mutant. Neither the HA-tag, nor the Δ CID mutant, interfered with the inhibitory effect of T9 on CLC-5 surface expression. Importantly, this demonstrates that pAC and CID operate independently on surface expression and ion transport, respectively. These results are now shown in the new Suppl. Fig. 7 and are mentioned in Results.

Fig. 3, compared to the effects of T9s on CLCs on the PM where direct electrophysiological recordings were used, those on the endosomes, the native environment of some of the CLCs, were less quantified by using vacuole generation instead of patch clamp recording. To minimize the uncertainty, the authors perhaps should also include data quantifying the the amount of CLC proteins on the organelle with and without T9s.

Your comment addresses a complex issue:

The amount of CLC-3 to CLC-5 proteins indeed depends strongly on their T9-subunits, as shown most impressively for T9a/T9A double KO fibroblasts (Fig. 1f). With single T9A or T9B KO in mice, or mice heterozygous KO for one, and homozygous for the other allele (Fig. 1e, now Fig 2b), the situation is more complex as both T9 isoforms can partially substitute for each other. Since all three CLCs localize very predominantly to endosomes, this means that the protein levels in those compartments will be affected to almost exactly the same degree.

Conversely, we also observed ‘stabilization’ of CLC proteins by T9 proteins in overexpression. The extent of this ‘stabilization’ depended on the particular CLC, with CLC-4 being the most sensitive (because of its need for CLC-3 to exits the ER?). These changes were often accompanied by changes in MW, suggesting differences in glycosylation due to changed ER-Golgi trafficking. The mechanism of the ‘stabilization’ likely involves several processes (changed trafficking, ERAD, lysosomal degradation). We prefer to not show such difficult-to-interpret data at this point. Nailing down the exact mechanisms of this ‘stabilization’ will be very challenging, would require much more work and is clearly beyond the scope of our work.

Likewise, the quantification of protein levels on specific endosomal compartments is extremely challenging and the interpretation of our results do not depend on such a quantification. Since purification of endosomal compartments is difficult and never absolute, and may be influenced by the ‘swelling’ of endosomal compartments by overactive CLCs, we prefer to look at total protein levels in Western blots. Immunocytochemistry has its own problems in quantification.

9) it'd be useful to show data to quantify the embryonic lethality by simply including the number of double homozygous embryos found in E12.5

We quantified the number of embryos from different genotypes at E12.5. From crossing double heterozygote mice, we obtained a total of 98 embryos of which 6 were double KO, 6 WT, 4 T9A-KO and 7 B-KO. We theoretically expect 6,25% of each – thus, there is no evidence for embryonic lethality before E12.5. Since we observed no live double KO offspring after birth, we assumed that the embryos die after E12.5. However, we cannot strictly exclude that $T9a^{-/-}/T9b^{-/-}$ mice die immediately after birth and are eaten by their mothers. We are now more cautious in the abstract 'and died embryonically or shortly after birth' and deleted 'embryonic' in results 'with embryonic lethality of double KO mice occurring after E12.5.'

10) Methods/Knockout generation: how many independent founders were used and how many, if any, generations of backcrossing were performed before experimental use?

We obtained several founders for T9A KO mice and initially tested two founders for the abundance of CICs in brain and kidneys, with identical results. In the end, we expanded only one of the lines. For T9B we also obtained several founders but only one had a deletion large enough to be easily detected by PCR during genotyping, so only one founder has been used.

The mutations were introduced by injection in fertilized in C57BL/6N oocytes. Mice were bred for a minimum of three generations with WT C57BL/6N before establishing the line.

We now mention these details in Methods.

Reviewer #2 (Remarks to the Author):

In this highly interesting manuscript, the authors have characterized the proteins TMEM9A and B (T9A/B) as β -subunits of the vesicular CLC transporters (vCLCs) CIC3, 4 and 5 and defined their role in trafficking and the regulation of transport. In a comprehensive set of conclusive experiments, they have initially identified T9A/B as co-migrating proteins with vCLCs in brain extracts by complexome profiling and multi-epitope affinity purification. They have also shown that mice containing a knockout of either T9A or B are viable while a double knockout leads to embryonic mortality, suggesting that these proteins are essential but can to some extent replace each other. In immunocytochemistry experiments it was shown that T9A/B target vCLCs to endosomes and that this targeting requires an acidic cluster at the C-terminus. Besides its role in trafficking the authors also showed that T9A/B inhibits the transport function vCLCs via an interaction with their C-terminus. Finally, previously described disease-causing mutations of vCLCs were found to interfere with the interaction with T9A/B. The study is accompanied by a complementary manuscript which provides a structural explanation for the described properties. Although the role of T9B as regulator of CIC3 and 4 was recently described, this work provides comprehensive functional and cell biological insight that goes far beyond the previous study. I thus consider this work as strong candidate for publication in Nature Communication with only minor modifications.

We thank the reviewer for appreciating the novelty and impact of our work.

I have the following (minor) remarks the authors should address:

- Fig. 1b, It is not clear what the axes refer to.

Thank you for pointing this out – indeed, while colleagues working with single-cell sequencing will be familiar with clustering data as done in the present t-SNE analysis, we assume that many others may not understand it. Dim1 and Dim2 are virtual dimensions which result from clustering multidimensional data into two dimensions using the t-SNE algorithm, as described in Methods. In Results, we now write: ‘Multi-epitope affinity-purifications from mouse brain combined with mass spectrometry (meAP-MS) confirmed effective co-assembly of the TMEM9 proteins with CIC-3 to -5, indicated by their tight co-clustering in two-dimensional t-SNE plots²²’, with reference 22 describing the method.

- Figure 1 is extensive and could be split into two figures.

Thank you for this useful suggestion. We have now put former panels 1d-k into a new Fig.2. This gave us also more space in each figure, which we have used to

(a) include in Fig. 1 not only the previous panels a-c, but also both panel of the previous Extended Data Figure 1 (t-SNE plot for T9B and T9A co-IP with various CLCs) and include schematic diagrams for the chimeras as suggested by reviewer 1,

(b) include images showing antibody specificities (using kidney from KO animals) together with previous panels d-k in the new Fig. 2.

- Line 85: Renal expression of CIC5 is dependent on T9B

We have changed the wording to ‘Renal expression of CIC-5 is dependent on T9B’, which the reviewer prefers over our previous ‘Renal expression of CIC-5 depended on T9B’.

- What is meant with CIC-4 needs CIC-3 for stability? (line 92)

*We had shown in Weinert et al., EMBO J 2020 (PMID: 32118314), that the levels of CIC-4 protein is markedly reduced in tissues of *Clcn3*^{-/-} mice. This was not accompanied by changes in *Clcn4* mRNA, suggesting that CIC-4 needs CIC-3 for protein stability. We further showed that CIC-4 needs CIC-3 for its exit from the ER, suggesting increased ERAD as a mechanism for CIC-4 degradation in the absence of CIC-3.*

*We have now expanded our sentence to: ‘It may be explained by the concomitant reduction of CIC-4, which is observed in brain and other tissues of *Clcn3*^{-/-} mice²⁹, and by low CIC-5 levels in brain.’ to be more explicit.*

- Line 106: Please state explicitly for which compartments rab2 and lamp2 serve as markers.

*We used *rab5* and *lamp2* as markers for endosomes and lysosomes, respectively, and now write: ‘..... sparsely co-localized with early endosomal *rab5* or late endosomal and lysosomal *lamp2* ...’*

What is the conclusion of the poor colocalization of TMEM9A/B with both markers?

The sparse colocalization of T9 proteins with the endosomal markers is probably owed to shielding of the epitope recognized by the T9 antibodies when in complex with the CLC proteins, which we describe at a later point in the manuscript. We added the following sentence to results: ‘The sparse co-localization might be owed, in part, to a shielding of T9 epitopes in CLC/T9 complexes (see below).’

- The conclusion stated in line 234-236 is not immediately clear and could be better explained

*With several disease-causing mutations in *CLCN4* we observed marked effects on vesicle size when inserted into CIC-3, but no effect on PM currents when inserted into CIC-5. As argued in the discussion, this indicates that the vesicle enlargement assay is much more sensitive than the PM current assay, the latter requiring a strong relief from T9-mediated inhibition to give currents above background PM currents. In comparison to mutations totally abolishing inhibition by T9, e.g. deletion of the C-terminus, the disease-causing mutations only partially weaken the inhibition, sufficient to be detected in the vesicle assay (and to result in disease), but still far from total disinhibition.*

*We have reworded to: ‘Hence the degree of disinhibition of ion transport by pathogenic *CLCN4* R652T and I655V mutants and their CIC-3 and CIC-5 equivalents is incomplete. However, this partial activation of CLC/T9 ion transport suffices to enlarge vesicles and to cause disease.’*

- It would be helpful if the authors refer to figures in the discussion to clarify their arguments.

Thank you for this suggestion. We have inserted references to Figures at many places in the Discussion.

- Although the role of the TM helix of T9A/B in the interaction with vCLCs was demonstrated here, the length of the interaction region referred to in line 300-301 was shown conclusively in the accompanying manuscript.

We have now referred to the accompanying paper, which is now under revision in NSMB, in: ‘The length of the CLC/T9 interface, as deduced here from functional studies and later confirmed by cryo-EM structures⁴⁶, suggests that posttranslational modification may synergistically activate vCLCs’

- It would be helpful if the authors would include a discussion figure that schematically summarizes the conclusions of the work.

We have designed a new Fig. 7 that shows some of the main conclusions of our work in a graphical manner. We think it will be very helpful.